# Impact of vaccination timing and coverage on measles near elimination dynamics: a mathematical modelling analysis

Anne M. Suffel [1,2] ✉, Charlotte Warren-Gash [1], Helen I. McDonald[2,3], Adam Kucharski[1,4] & Alexis Robert [1,4]

The Joint Committee for Vaccination and Immunisation recommended to implement an earlier second dose for the Measles-Mumps-Rubella (MMR) vaccine starting in 2026 in the United Kingdom. We investigated the impact of these changes on measles transmission in England. Using an age- and region-stratified mathematical model, we simulated outbreaks with different vacci-nation schedules and coverage, using electronic health records and outbreak data from 2010 to 2019. Delivering the second MMR dose at 24 months reduced cases by 11.86% (IQR: −3.3; 23.81%) compared to the current schedule (3 years and 4 months) and showed a 22.39% (IQR: 10.05; 32.79%) reduction of cases if achieving the same coverage as the first MMR dose. The effect of delivering an earlier second MMR was lower when waning of vaccine-induced immunity was included (5.28% (−10.92; 19.63%)). Increasing first-dose coverage by 0.5% annually yielded slightly better outcomes than an earlier second dose (14.68% reduction, IQR:1.19; 27.49.9%). While improving first-dose uptake had the greatest impact, it may be difficult to achieve. Thus, an earlier second MMR dose can be a feasible alternative to reduce the measles burden in England where measles transmission follows typical near-elimination dynamics.

Measles is a highly contagious disease that can lead to severe illness affecting almost every organ system[1]. However, measles infection is prevented very effectively by vaccination. The global eradication of measles through vaccination is feasible and several countries have already achieved measles elimination[2]. The first measles vaccine was developed in 1963[3,4], and many countries now use the Measles-Mumps-Rubella (MMR) vaccine. A single dose of measles-containing vaccine was shown to be at least 95% effective in preventing clinical measles in preschool children[5,6]. The first dose of vaccine (MMR1) should be administered around the first birthday in countries with low incidence[7], as earlier first dose vaccination can reduce the vaccine effectiveness[8]. A second dose (MMR2) is routinely recommended as not all children respond to MMR1[3]. The guidance on timing of MMR2 varies across countries and is recommended by the World Health Organization between the second year of life and school entry[3]: Some European countries (e.g. France, Germany) administer MMR2 before the second year of life, while others (e.g. Finland, Latvia or Poland) only recommend MMR2 around school entry between the ages of 5–7 years[9]. Since 1996, the vaccine schedule recommended by the National Health Service in the United Kingdom states that MMR should be given at the ages of 1 year and at 3 years and 4 months[10,11]. It remains unclear how these different vaccination strategies impact measles dynamics in low-incidence settings such as the European region.

Since measles is highly infectious, high vaccine coverage is required to mitigate the risks of outbreaks—several studies estimated that a coverage of around 95%[12,13], achieved by the age of five, was necessary to ensure elimination of measles transmission[14]. Uptake of the MMR1 vaccine by age 2 in England declined in the late 90s until the

[1]Faculty of Epidemiology and Population Health, London School of Hygiene and Tropical Medicine, London, UK. [2]NIHR Health Protection Research Unit in Immunisation at London School of Hygiene and Tropical Medicine, London, UK. [3]Faculty of Life Sciences, University of Bath, Bath, UK. [4]Centre for Mathe-matical Modelling of Infectious Diseases (CMMID) at the London School of Hygiene and Tropical Medicine, London, UK. ✉e-mail: Anne.suffel@lshtm.ac.uk

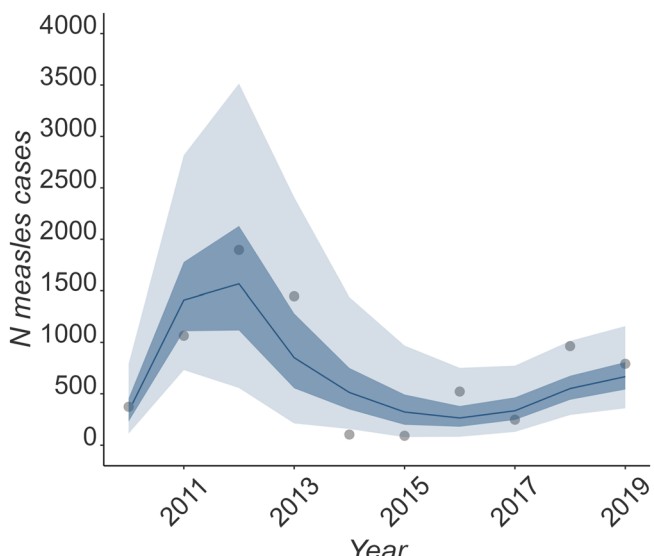

**Fig. 1 | Number of measles cases in comparison to the simulation cases by the model.** Number of notified measles cases in England (dots) and median number, interquartile range (light blue shade) and 95% simulation interval (dark blue shade) of simulated measles cases by the transmission model (blue line).

early 2000s, ultimately reaching 80%, and then increased again until 2015 with additional catch-up campaigns[15]. However, the coverage has now been declining since 2015[16] and was additionally impacted by the COVID-19 pandemic[17,18]. These variations have been associated with several large outbreaks (above 1000 annual cases) in the UK since 2010[19–21].

The UK Joint Committee on Vaccination and Immunisation (JCVI) published a recommendation to bring MMR2 forward to the age of 18 months by 2026 in order to improve coverage[22]. This decision was based on a study by Lacy et al. indicating that an earlier vaccination schedule in some London boroughs led to 3.3% (95% CI: 1.3–5.3%) higher uptake in London at the age of five[23]. However, it remains uncertain how changes in the vaccination schedule would influence vaccine uptake in the long term. In particular, shifting MMR2 to the age of 18 months would require an additional vaccination appointment, which could lead to a lower acceptability or might be difficult to organise for parents, which might lead to lower coverage[24]. Transmission models have been shown as a useful tool for policy recommendations regarding vaccination schedules if applied correctly to the local epidemiology of measles[25].

We aimed to quantify the impact of different measles vaccination strategies on the risk of measles outbreaks in England, including the existing and new vaccination schedules in the United Kingdom and alternative schedules from other European countries[19].

## Results

In the reference scenario, the stochastic simulations generated with the model replicated the overall pattern of measles outbreaks in England between 2010 and 2019, capturing the peak of cases in 2012 and a second smaller peak in 2018 (see Fig. 1). The simulations also replicated the age distribution of the cases observed in the data (Supplementary Figs. S5 and S6). Predicted measles cases in a majority of simulations in 2013, 2016 and 2018 were below the observed outbreak data, and above the observed cases in 2014 and 2015 (age and spatial distributions are available in the Supplementary Tables S5 and S6).

In the reference scenario, the transmission model simulated across 2500 simulations showed a median of 7169 cases (IQR: 6096; 8534.5, see Table S7 in the Supplementary) between 2010 and 2019 (7502 were reported in the data).

### Changes in coverage

Improving the annual MMR1 coverage had a strong impact on reducing measles cases if achieved on a sufficient scale. A 0.25% improvement of coverage for MMR1 led to a small reduction of cases (8.44% median difference, IQR: −7.96; 21.91%), a 0.5% improvement led to a 14.68% reduction (IQR: 1.19; 27.49%) of cases and an improvement by 1% led to 28.39% averted cases (IQR: 17.53; 37.68%, see Table S4 and Fig. 2). Improvements of MMR2 coverage of up to 3% by year only had a limited impact on the number of measles cases (see Fig. 2).

### Changes in timing

Moving MMR2 to 2 years instead of 3 years and 4 months reduced the overall number of cases by 11.86% (IQR: −2.90; 23.35%) in comparison to the median number of cases in the reference scenarios (see Fig. 3). These reductions are slightly smaller than the impact of increasing MMR1 coverage by 0.5% (see Table S4), and this effect was slightly lower if it was associated with a reduction of MMR2 uptake (7.69% decrease [−8.72; 21.11]% if MMR2 uptake decreased by 3%, see Fig. 3C–F, Table S4). If the coverage of the earlier MMR2 was similar to MMR1, up to 22.38% (IQR: 10.05; 32.79%) of cases were averted.

Delaying MMR2 at the school entry age of 5 performed worse than all the other vaccination schedule scenarios: the number of simulated measles cases increased by 73.86% (IQR: −114.32; −41.69%).

### Changes in the age structure of measles cases

The different vaccination strategies had an influence on the age distribution of measles cases.

MMR2 given at school age increased measles cases across all age groups and almost doubled the number of cases in children aged between 4 and 6 in comparison to the reference scenario (see Fig. 4C, F). An earlier delivery of MMR2 led to a lower proportion of measles cases in children aged between 2 and 4. An MMR1 uptake of 0.5% decreased the number of measles cases across all age groups (see Fig. 4).

Through either increasing the coverage of MMR1 or advancing MMR2, the years with the highest proportion of cases averted are 2012, 2013, 2014 and 2015 (see Table S5). When increasing MMR1 by 0.5% or advancing MMR2, the highest proportion of avoided cases was in the North West of England, Yorkshire and the Humber and the North East (see Table S6). However, the differences in proportion of cases averted per region were small in both scenarios, with a median proportion ranging between 14.09% and 16.24%.

### Sensitivity analysis

When using Cover of Vaccination Evaluated Rapidly (COVER) data instead of Clinical Practice Research Datalink (CPRD) data, there was a lower reduction in cases with a 7.21% (IQR: −11.31; 22.11%) averted cases when MMR2 was given at age 2 in comparison to the median of the reference scenario (see Fig. S7 and Tables S7–S9). Including waning (starting from 5 years of age) to the model also led to a lower median reduction of cases than the reference scenario (5.28% reduction, IQR: −10.92; 19.63%; see Figs. S9 and S10 and Table S10–S12).

Changing the start of waning to 3 years of age (with the same waning rate as the previous waning scenario) also had an impact on the reduction in case numbers (3.79% reduction for earlier MMR2, IQR: −12.93;18.83%, see Tables S9 and S10, Figs. S9 and S10). In the short and medium term, waning did not fully cancel out the impact of moving the age of delivery of MMR2.

## Discussion

We explored how different vaccination schedules could impact the dynamics of measles outbreaks, and showed that advancing the second dose of MMR could result in a reduced number of measles cases even if the uptake did not improve as hoped by the policy recommendation from JCVI[22]. We showed that bringing MMR2 forward could

lead to a reduction in case numbers slightly smaller than increasing the MMR1 coverage in each region by 0.5% over 10 years. In the reference analysis, both resulted in an 11.86–14.68% reduction of cases (IQR for 0.5% increase MMR1: 1.19; 27.49%; IQR for early MMR2: −2,90; 23.35%) between 2010 and 2019. These changes in the vaccine schedule could lead to significant short-term improvements and may mitigate risks of large outbreaks. However, only a smaller decrease in cases of 4-8% was observed if an earlier administration of MMR2 was accompanied by a reduction in coverage between 3 and 5%. If an earlier MMR2 was taken up with a comparable coverage to MMR1, a greater reduction of cases could be achieved than increasing MMR1 by 0.5% across all sensitivity analyses (22.38% (IQR: 10.05; 32.79) reduction in the reference scenario).

A previous study has shown that the implementation of accelerated MMR immunisation schedules in some boroughs of London[23] resulted in an increase of vaccine coverage, although the overall coverage remained lower than in the rest of the country[26,27]. However, this only works under the assumption that an additional vaccination appointment will be accepted by parents as the MMR2 is currently given at the same time as the 4-in-1 preschool booster. Determinants of childhood vaccine uptake are complex, and include parental vaccine confidence, opportunities to ask questions, reminders and barriers to vaccination appointments, which could all be impacted by changes in vaccine schedules[28]. However, vaccine uptake is generally higher for the earlier appointments in the UK childhood vaccination schedule and so it may be reasonable to assume that an earlier MMR2 date is likely to result in higher uptake[27]. Introducing an earlier second MMR2 at a similar time to a routine health visit for the child might also offer an additional opportunity to remind parents of the vaccination and provide information.

Although moving MMR2 to the age of five resulted in more cases than other scenarios, there are pragmatic factors which might make this option favourable: The administration of a vaccine in a school setting is convenient for families and offers peer support and opportunities for education related to personal health and health care[29,30]. However, we previously observed lower vaccine uptake for appointments scheduled later in the routine immunisation schedule[27].

Improving the MMR2 coverage alone did not influence outbreak dynamics in the reference scenario. This suggests that in order to mitigate the risk of outbreaks, it is most important to focus on unvaccinated individuals where possible.

The period of the simulation covered only 10 years and does not account for the long-term effects of changing the age of vaccine receipt. This might be especially important as recent evidence suggested a potential slow waning of immunity from the MMR vaccine[31,32]. A second vaccine dose given at a younger age might lead to marginally less protection in adults in the long term in a non-endemic setting[31,33,34]. In a sensitivity analysis, we showed that the median number of cases with early MMR2 vaccination remained 4% smaller than the reference simulations when waning started at 3 years instead of 5. This shows that the short and medium-term consequences of waning might decrease the impact of early MMR2 vaccination in our model. Close monitoring of transmission in vaccinated adults is essential to understand long-term immunity of vaccinated individuals in countries near elimination.

In the reference scenario without waning, the risk of primary vaccine failure estimated when fitting the model to the case data is 5.2% (95% Credible Interval (CI): 4.9%; 5.5%). In the early MMR2 scenario with no change in coverage, most vaccinated children aged 2 and 3 have received 2 doses of vaccine, leading to a reduction of vaccinated children with primary vaccine failure. This increased immunity in children aged 2 and 3 causes indirect protection to the rest of the population, leading to an overall 11.86% reduction of cases compared to the median number of cases in the reference simulations (IQR:

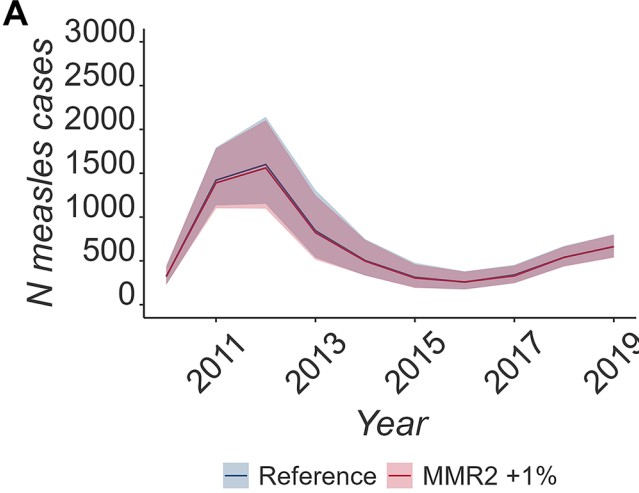

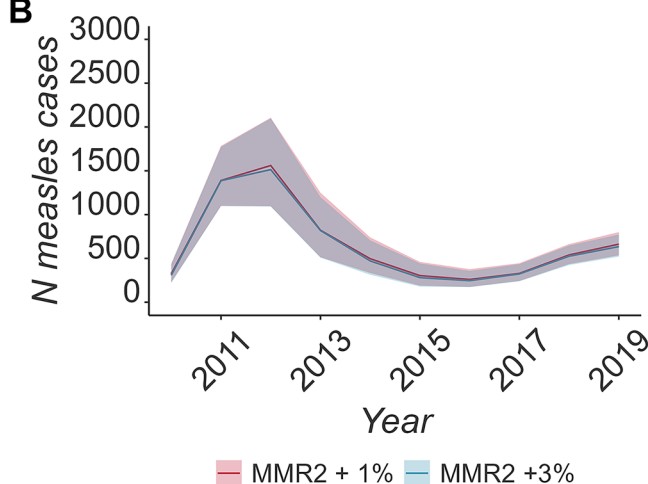

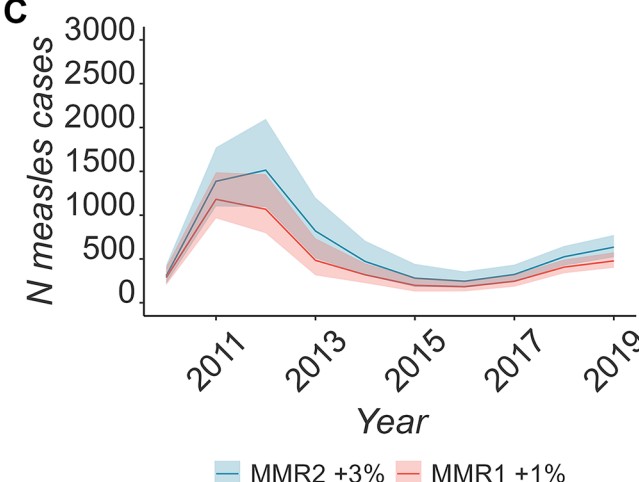

**Fig. 2 | Comparison of the reference scenario to scenarios improving vaccination coverage.** Comparing the median number and IQR (shaded area) of simulated cases across simulations between **A** Reference scenario against MMR2 increased by 1%, **B** MMR2 increased by 1% against MMR2 increased by 3%, **C** MMR2 increased by 3% against an increase of MMR1 by 1%.

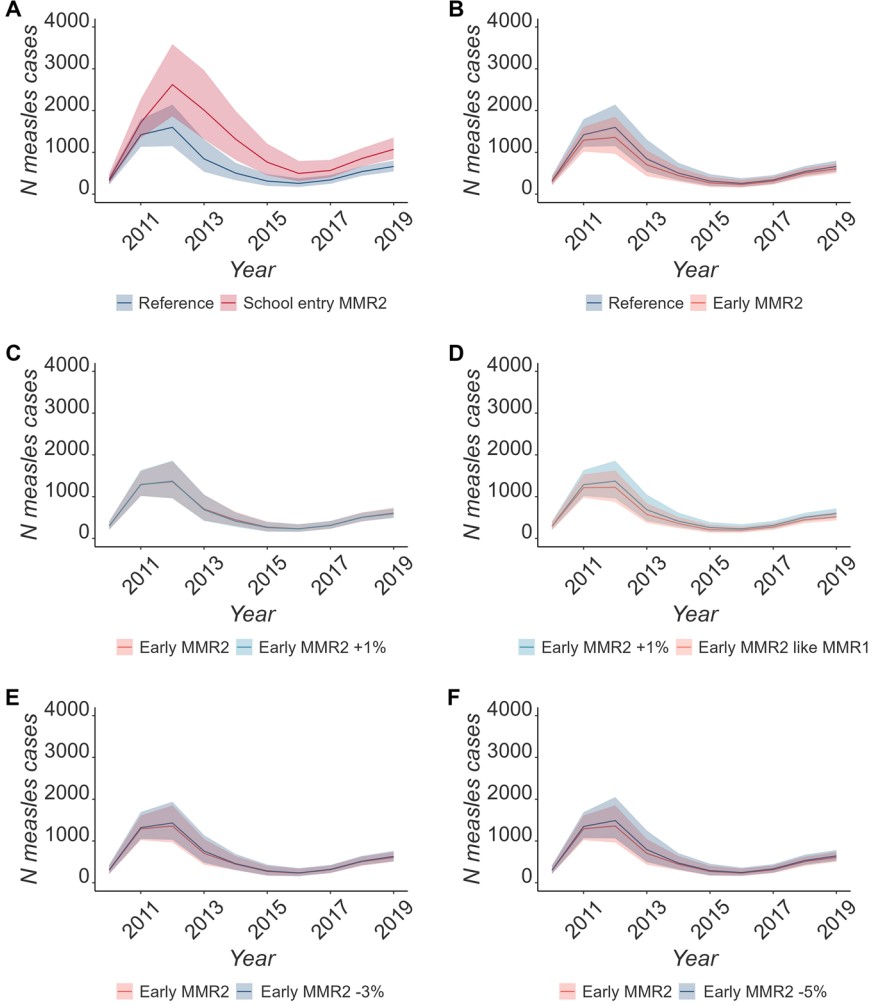

**Fig. 3 | Comparing the reference scenario to scenarios with changes in the immunisation schedule.** Comparing the median number and IQR (shaded area) of simulated cases across simulations using CPRD data between **A** Reference scenario and MM2 given at school age, **B** Reference scenario and MMR2 given at the age of two, **C** MMR2 given at the age two against an increase of MMR1 by 1%, **D** increased MMR1 by 1% and an earlier MMR2 with the same coverage as MMR1. **E**, **F** are comparing the early MMR2 with the same uptake as before against a drop in coverage by 3% and 5% respectively.

−2.90% to 23.35%). When waning is included in the model, the risk of primary vaccine failure estimated by the model is lower (2.5%, 95% CI 2.2%; 2.9%), and the impact of early MMR2 on overall case number is also lower (5.28% (−10.92; 19.63)). It is difficult to assess which model is closer to the true impact of early MMR2 as both reference simulation sets slightly diverge from the data: the number of measles cases aged 2 and 3 was underestimated in the model with waning (315 (IQR: 266–373) cases, while 486 were observed in the data), hence it might underestimate the protective effect of bringing the second dose forward. The model without waning overestimated the number of vaccinated cases aged 2 and 3 (164 (IQR: 140; 193) cases, while 52 were observed in the data); hence, it might overestimate the protective effect of an earlier second dose.

More work is needed to understand how vaccination schedules influence outbreak risks in near elimination countries: Vaccination schedules differ between near elimination countries, but differences in historical coverage, previous incidence, age structure of the population, and spatial distribution of coverage make direct comparison challenging. With the increase in measles cases observed across most European countries in 2023 and 2024[35], it is important to understand whether the early MMR2 schedule is consistently associated with lower outbreak risk[36]. Further epidemiological or mathematical modelling analysis using historical case and vaccination data from near-

elimination countries would be needed to fully understand how our findings can be transferred to other settings.

Strengths of this study included the use of a compartmental model estimating the intensity of transmission. This approach allowed us to test different hypothetical scenarios in detail, which would not be possible in real life due to financial and logistical constraints. We also used detailed, representative data on vaccination uptake, which allowed us to study changes in uptake by age more precisely.

There are several limitations to this approach. Firstly, although the model is stratified by region, outbreaks in low-incidence settings are usually driven by heterogeneity in vaccination in certain groups and communities that cannot be captured on a regional scale[33,37]. Increasing the spatial granularity of the model would mean using upper-tier local authorities instead of regions, and would lead to two main issues: Firstly, CPRD vaccine data was not available on a more granular scale to protect anonymity. Secondly, increasing spatial granularity would multiply the number of strata in the model: there are nine regions in England, but more than 100 local authorities, which would greatly increase the number of compartments in the model (100 spatial units, 12 age groups and 12 compartments per strata would result in more than 10,000 compartments). This would slow down fitting the model and generating the stochastic simulations.

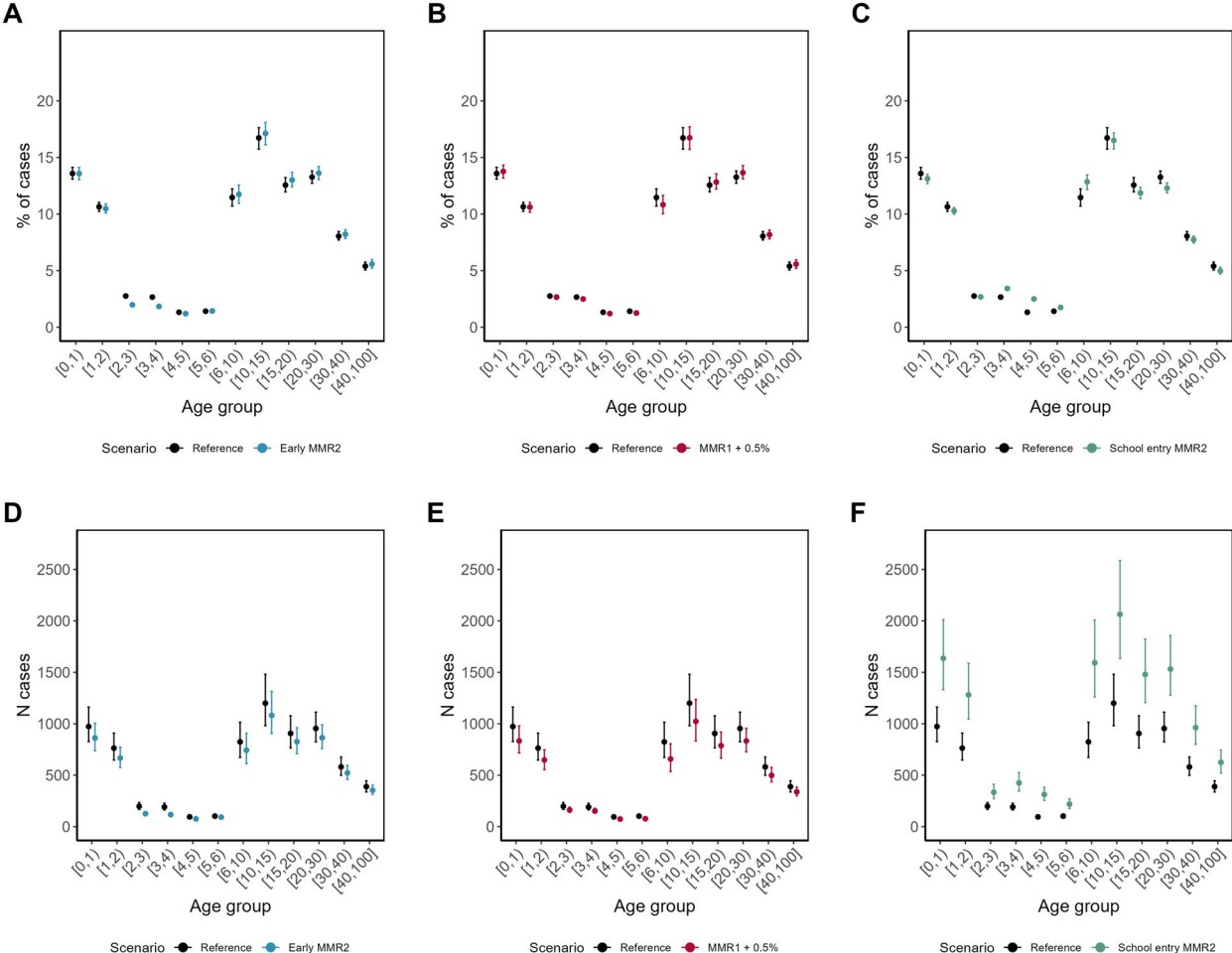

**Fig. 4 | Age distribution of measles cases by different vaccination scenarios.**
Comparison of measles cases per age compartment as proportion of all cases and
absolute numbers of cases between the reference scenario and an earlier MMR2
(**A**, **D**), between the reference scenario and MMR1 improved by 0.5% (**B**, **E**) and
MMR2 given at the age of five (**C**, **F**). The error bars represent the interquartile range
across all 2500 simulations per scenario.

Secondly, we assumed an immediate implementation and
acceptance of the altered vaccination schedules. However, it is likely
that there would be a transition period from one schedule to another;
hence, our model may overestimate the number of cases avoided at
the start of the transition. Thirdly, due to the use of 1-year age bands in
the model, we modelled a schedule change moving MMR2 forward to
the age of 2 years instead of the suggested 18 months from JCVI. The
reduction of cases by bringing MMR2 to 18 months forward might
reduce the number of cases even more than in the results presented.
Finally, this analysis focuses on counterfactual scenarios of measles
transmission between 2010 and 2019, and does not account for the
impact of vaccination trends, demographics, or global measles
dynamics on future outbreaks in England.

For policymakers, this work provided insight into different trade-
offs when making decisions related to vaccination. While we found that
increasing MMR1 coverage would be most effective at reducing cases,
this may be challenging as interventions to improve vaccine uptake and
reach unvaccinated individuals are usually very context-specific and
have to be tailored to vulnerable groups in the population, with mixed
evidence for general interventions such as easier access and reminders[38].
The COVID-19 pandemic increased vaccine hesitancy in parents towards
general childhood immunisations[39], which presents an additional chal-
lenge for improving coverage. Bringing the second dose forward to the
age of two or younger could improve uptake at a younger age, as
childhood immunisations show better uptake the earlier that they are
recommended in the immunisation schedule[27]. However, some of the
benefits of an early MMR2 might be cancelled out by waning of immu-
nity after an earlier vaccination, and will require close monitoring. With
the appointment at 3 years and 4 months still in place for the preschool
booster, this strategy would still provide an additional opportunity for
health care contact with children and their parents to catch up on an
MMR dose, which may have been missed before.

## Methods
### Study design
We simulated outbreak risk under different vaccination strategies and
coverage between 2010 and 2019. We chose this period to avoid the
potential disruptions of routine immunisation programmes associated
with the COVID-19 pandemic. We generated stochastic simulations
using a mechanistic transmission model that had previously been fit-
ted to the daily number of confirmed cases reported in England stra-
tified by age, region, and vaccination status[31]. We used the parameter
estimates obtained from the deterministic model fits to simulate sto-
chastic outbreaks between 2010 and 2019. The modelled scenarios
included vaccination schedules from other European countries, such
as introducing MMR2 at an earlier age, or recommending a later MMR2
at school entry, improving coverage of MMR1 and MMR2 separately,
and changes of schedule and coverage together.

### Outbreak data
Data on all laboratory-confirmed measles cases in England between
2010 and 2019 were obtained from Public Health England (now UK

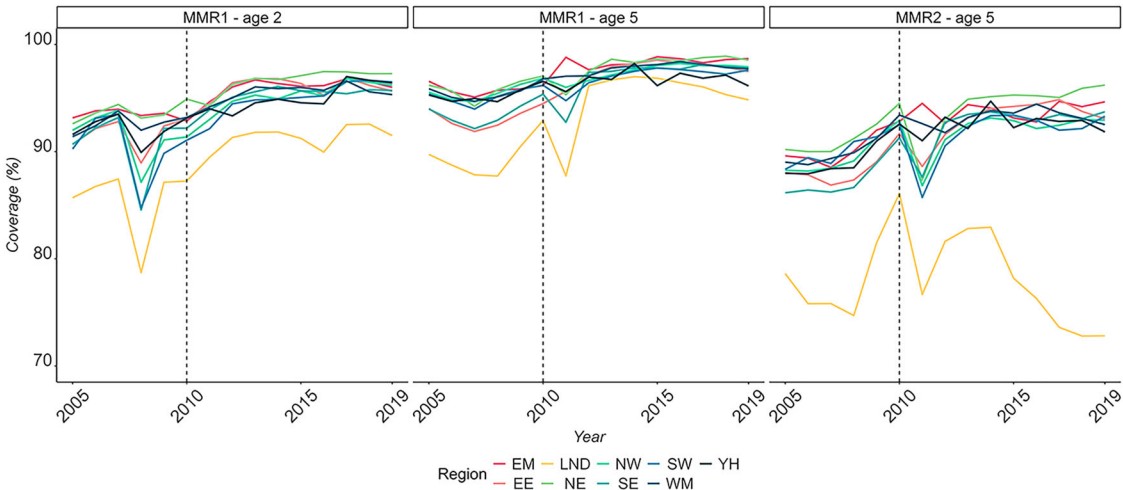

**Fig. 5 | MMR vaccination coverage by region.** Vaccination coverage estimates stratified by region for the MMR1 at ages 2 and 5 years, and for MMR2 at the age of 5. EM East Midlands, LND London, NW North West, SW South West, YH Yorkshire and the Humber, EE East of England, NE North East, SE South East, WM West Midlands.

Health Security Agency). This dataset included the date of onset, region of residence, age and vaccination status of the 7504 cases and was used to fit the transmission model (more details on the dataset in Supplementary Section S1).

**Vaccine data**

Two data sources of vaccine coverage were used: The CPRD Aurum to estimate vaccine coverage by region and 1-year age bands and the COVER to supplement missing data for age groups not included in CPRD Aurum.

CPRD Aurum is a primary care dataset from general practitioner (GP) surgeries using EMIS Web® software that contains patient-level information on symptoms and diagnoses, clinical tests and results, immunisations, prescriptions and referrals to other services[40]. In 2022, CPRD Aurum contained data from around 25 million patients and was broadly representative of England by geographical spread, age, sex and ethnicity[41]. Using a validated algorithm to identify vaccination records[42], vaccination coverage at the ages of 1, 2, 3, 4, and 5 years stratified by region was estimated from the electronic health records. The results were previously published[42]. The CPRD data were only available for children born between 2006 and 2015; thus, it covered all age bands between 0 and 5 in the years 2010 to 2016.

COVER is a dataset published by NHS Digital summarising UK vaccination coverage at the ages 2 and 5 for the MMR vaccine for England and by geographical region[26]. The COVER national coverage was available for children born between 2000 and 2019. The region-stratified coverage was only available for children born in 2004 and after.

For children born before 2006 or after 2015, the CPRD vaccine data had to be supplemented with estimated values from COVER data. COVER uses aggregated GP information based on operational data, which may be incomplete, not fully representative and not quality assured[43]. COVER data that has been corrected for underascertainment is closer to CPRD estimates. A comparison between COVER data and CPRD data can be found in the Supplementary Material (see Supplementary Section S2). Based on this, COVER estimates used to supplement missing CPRD data were adjusted using the assumption that 50% of the unvaccinated children were vaccinated but did not have their vaccine recorded[15]. These corrected values were consistent with estimates from previous studies[27]. To estimate the vaccine coverage in the missing age bands in the COVER data, we applied the relative difference of the age bands for the last completely available years, i.e. 2006 and 2017, to the COVER estimates to supplement the values for ages three and four. All values of vaccine coverage were stratified by region (see Fig. 5).

In a sensitivity analysis, we fitted the model to the uncorrected COVER data for which the missing age strata were supplemented by the proportional change in uptake between age groups as observed in the CPRD data.

As the recommended ages for MMR vaccination are 1 year (MMR1) and 3 years and 4 months (MMR2), most children receive a first dose between one and two, and a second dose between three and four. Therefore, the vaccination coverage for MMR1 at age 1 and MMR2 at age 3 is nearly zero in the data, but a large proportion of children aged one have actually received a dose of vaccine (same with MMR2 for children aged three). To adjust the coverage data in the 1-year age band structure of the model, we assumed the coverage of MMR1 at the age of 1 was 75% of the coverage for MM1 at age 2, and 50% of the coverage at age 3, of the coverage of MMR2 at age 4.

**Transmission model**

We used a compartmental transmission model with compartments for susceptible, exposed, infected and recovered individuals (SEIR model) and single and double vaccinated individuals to reproduce the measles dynamics observed in England by age group, region, and vaccine status. The transmission model was presented in detail in a previous publication[31]. A more detailed description of the model fitting process and model parameters can be found in the Supplementary Material (see S3). The parameters estimated by the model were then used to generate stochastic simulations, describing the case dynamics simulated under a range of alternative scenarios (see Fig. 6)[31].

We ran 2500 simulations per vaccination strategy by drawing 100 parameter sets from the model fits and running 25 simulations per parameter set. Parameters drawn from the model fits included infection rate, duration of maternal immunity, parameters for seasonality of transmission and importation, report rate of imported cases, vaccine effectiveness, existing immunity in older generations and parameters of spatial spread (see Table S7). In the stochastic simulations, the number of transitions was computed using binomial draws, with a rate of transition derived from the distribution of the population between compartments at the previous time step and the parameter set. Counterfactual scenarios were matched by using the same seed for the stochastic simulations.

**Alternative vaccination scenarios**

We explored how changes in vaccination impacted the number of cases simulated by the transmission model. To do so, we generated sets of stochastic simulations using the same parameter sets as the reference simulation set (described in section S2 in the supplementary

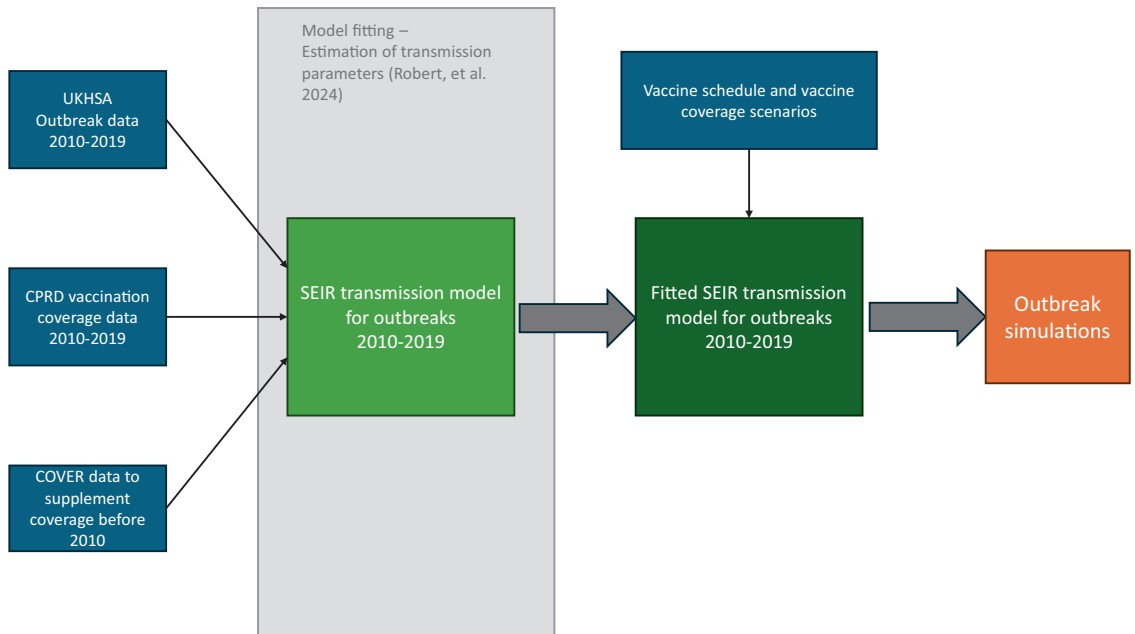

**Fig. 6 | Overview of the analyses.** Process of fitting the compartmental transmission model and generating the outbreak simulations based on different vaccination scenarios. UKHSA UK Health Security Agency, CPRD Clinical Practice Research Datalink, COVER Cover of Vaccination Evaluated Rapidly. The transmission model was stratified by region, age groups and vaccine status. The model includes 12 age groups: <1, 1–2, 2–3, 3–4, 4–5, 5–6, 6–10, 10–15, 15–20, 20–30, 30–40, 40+.

material), but with changes in vaccination schedule, vaccine coverage, or both. Changes in schedule were based on other vaccination schedules implemented in Europe. Table 1 summarises the different scenarios for which we simulated measles outbreaks in England.

We used the median number of cases between 2010 and 2019 and the IQR in each simulation set to estimate the impact of changes in vaccination schedule and coverage. The number of cases is compared to the reference set of simulations by computing the median and IQR of the percentage of change (increase or decrease) between each simulation set and the median number of cases in the reference simulations.

#### Changing vaccine coverage
We created scenarios in which the timing of MMR2 was not changed from the original schedule, but the overall coverage of MMR1 or MMR2 was increased across all regions and years between 2010 and 2019. We implemented an absolute increase of 0.5 and 1% for either MMR1 or MMR2 in every region between 2010 and 2019. As coverage for MMR2 at the age of five is lower than for MMR1, we further included scenarios increasing or reducing MMR2 coverage by up to 3%.

#### Changing MMR2 schedule
We explored two alternative scenarios with MMR2 recommended at the age of two instead of 3 years and 4 months. The first scenario assumed that the uptake would follow the same pattern of timeliness as the current MMR2 delivery; therefore, 1 year and 4 months were subtracted from all the dates when MMR2 was received. New coverage estimates were calculated from these updated dates. In the second, we assumed that MMR2 recommended at the age of two would be taken up with the same speed as MMR1, which is usually faster than MMR2[27]. In the new schedule recommended by the JCVI, MMR2 would be delivered at 18 months, but since our model is stratified by age groups (1-year age bands until 6), we used 24 months in the simulations.

We also explored the impact of moving the MMR2 recommendation to 5 years of age, assuming that the speed of uptake was similar to the reference MMR2 speed. New coverage estimates were calculated from the updated dates.

Finally, we looked at potential scenarios in which an earlier MMR2 would influence coverage for MMR2. For this, we explored an increase by 0.25, 0.5 and 1% and added a scenario in which the coverage for MMR1 and MMR2 were equal. Lastly, we added one scenario in which the coverage of MMR2 would be negatively impacted by bringing MMR2 forward in the schedule.

We assumed that all vaccinated children had received their second dose at 2 at the start of the simulations. This means that the effects of a transition period where part of the children would still be vaccinated on the original schedule are not considered in these scenarios.

#### Sensitivity analysis
We conducted several sensitivity analyses: (1) we recreated the scenarios with changes in the vaccination schedule using uncorrected COVER data to describe the annual vaccine coverage, with CPRD data informing the proportion of vaccinated children at 3 and 4 years of age (Section S5 in the Supplementary). (2) We included waning of vaccine-induced immunity from age 5, whereby vaccinated individuals who developed immunity upon vaccination can become infected, with infection risk increasing with age. In the model with waning, age is used as a proxy for time since vaccination, and the waning rate is based on estimates of our previous work[31]. (3) As moving MMR2 delivery to age 2 may influence when waning begins, we ran a sensitivity analysis with waning of immunity starting at age 3 when MMR2 was given early (Section S6 in the Supplementary)[31].

#### Inclusion and ethics
We received data governance approval from CPRD (protocol number 22_001706) and ethical approval from the London School of Hygiene and Tropical Medicine's research ethics committee (reference number 27651).

**Table 1 | Overview of the different vaccination strategies**

| Strategy name | Examples of European regions with this vaccination schedule in place[5a] | MMR2 schedule | Assumptions about vaccine coverage for MMR1 | Assumptions about vaccine coverage for MMR2 |
|---|---|---|---|---|
| Reference scenario | UK schedule 2024, Cyprus, Denmark, Ireland, Slovakia | MMR2 given at the age of 3 years and 4 months | Original coverage data for MMR1 | Original coverage data for MMR2 |
| Improving coverage of MMR1 | UK schedule 2014 | MMR2 given at the age of 3 years and 4 months | Annual MMR1 coverage improved by 0.5% or 1% at the age of 2 | Original coverage data for MMR2 |
| Improving coverage of MMR2 | UK schedule 2024 | MMR2 given at the age of 3 years and 4 months | Original coverage data for MMR1 | Annual MMR2 coverage improved by 1% or 3% |
| School Entry MMR2 | Belgium, Croatia, Finland, Latvia | MMR2 given at the age of school entry (age given) | Original coverage data for MMR1 | Original coverage data for MMR2 |
| Early MMR2 | France, Germany, UK schedule from 2025 | MMR2 given at the age of 2 years | Original coverage data for MMR1 | Annual MMR2 coverage improved by 0.25%, 0.5% or 5% Annual MMR2 coverage decreased by 3% or 5% |

[a]Some countries have wider ranges of recommended timings for the vaccine and this table does not reflect the exact timing in these countries.

## Reporting summary

Further information on research design is available in the Nature Portfolio Reporting Summary linked to this article.

## Data availability

The individual-level case data were collected by the UK Health Security Agency (UKHSA) and cannot be shared publicly due to privacy concerns. However, this data is not required to run the model and the simulations and was only used for the original model fitting presented in our previous publication, Robert et al.[31]. The individual-level vaccination data was obtained from the Clinical Practice Research Datalink (CPRD). CPRD does not allow the sharing of patient-level data to protect the patient's anonymity and privacy. The data specification for the CPRD dataset is available at: https://cprd.com/cprd-aurum-may-2022-dataset. For data access, a protocol needs to be submitted to the Electronic Research Applications Portal (eRAP) at www.erap.cprd.com and be approved for data access, which takes around 3 months. The COVER vaccination data is publicly available: https://www.england.nhs.uk/statistics/statistical-work-areas/child-immunisation/. The data to implement the alternative vaccination scenarios can be found in the GitHub repository: https://github.com/Eyedeet/measles_vaccination_scenarios/tree/main/Output.

## Code availability

The analysis code can be found in the following GitHub repository: https://github.com/Eyedeet/measles_vaccination_scenarios (https://doi.org/10.5281/zenodo.16812281) and used version v1.0.0 of the seir-vodin package (https://github.com/alxsrobert/measles_england_sir).

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

## Acknowledgements

A.S. and H.M. are funded by the National Institute for Health and Care Research (NIHR) Health Protection Research Unit in Vaccines and Immunisation (NIHR200929), a partnership between the UK Health Security Agency and the London School of Hygiene and Tropical Medicine. The views expressed are those of the author(s)

and not necessarily those of the NIHR, UK Health Security Agency or the Department of Health and Social Care. C.W.G. is supported by a Wellcome Career Development Award (225868/Z/22/Z). A.R. was supported by the National Institute for Health Research (NIHR200908), A.J.K. was supported by a Sir Henry Dale Fellowship jointly funded by the Wellcome Trust, the Royal Society (206250/Z/17/Z) and by the National Institute for Health and Care Research (NIHR200908). This work uses data provided by patients and collected by the NHS as part of their care and support (usemydata.org).

## Author contributions

A.S., C.W.G., H.M., A.K. and A.R. developed the analysis plan. A.S. and A.R. implemented the analysis, wrote the code, and ran the model. A.S. computed and collated the coverage data. A.S. and A.R. interpreted the results. A.S. wrote the first draft and the additional file. A.S., A.R., C.W.H., H.M. and A.K. contributed to the manuscript and read and approved the final version of the manuscript.

## Competing interests

The authors declare no competing interests.
