## [Transparent Peer Review file · Nature Communications]

Impact of vaccination timing and coverage on measles near elimination dynamics: a mathematical modelling analysis

Corresponding Author: Dr Anne Suffel

Version 0:

Reviewer comments:

Reviewer #1

(Remarks to the Author)

I read this paper with great interest as a valuable addition to the literature. However, I believe that revisions are needed prior to publication, as described in the following comments.

1. Lines 44-45: This sentence is not technically correct as written, since eradication has not been achieved. Suggest a rephrase such as, "The global eradication of measles through vaccination is feasible and elimination has been achieved in large geographical areas."
2. Lines 47-48: The sentence specifying 95% vaccine effectiveness is missing a citation. A citation for this sentence is essential.
3. Lines 48-50: I think it is somewhat misleading to say that MMR1 should be administered around the first birthday, since this is really the recommendation regarding mumps whereas this paper is focused on measles. The WHO recommendation is 9 or 12 months, minimum 6 months, for measles. More context and clarification is needed here.
4. Lines 123-132: I am not sure that I fully follow the logic here. If COVER data is not fully representative and not quality-assured, and you are correcting values based on CPRD data, wouldn't a scenario in which you extrapolate the CPRD data for the missing cohorts be valuable? It might be that this section needs further elucidation.
5. Reference #20: I went to check this paper referenced for the original transmission model, and it seems like the citation is incomplete in the bibliography.
6. Lines 150-152: Supplementary exhibit S5 does not clearly define what parameters were varied and under what assumptions. If 250 simulations per vaccination per vaccination strategy were run, it is important to include in this paper what elements could vary between each simulation.
7. Lines 187-189: If 1-year age bands are used and the model assumed 24 months rather than 18 months for the alternative scenario, how was the model able to account for a schedule of 3 years 4 months in the reference scenario? This needs to be clarified.
8. Section 2.5, Lines 160-202: In general, this entire section was quite difficult to follow. I might suggest revising Table 1 to more clearly delineate the different sets of assumptions for each scenario (e.g., a column for coverage, a column for MMR2 schedule), so that the text can more clearly link to the table and provide the additional information needed. The text in this section needs more clarity.
9. Lines 205-208: This comment connects to comment #4 above. If sensitivity analyses with alternative assumptions are included, I might suggest an additional scenario that only uses CPRD data – either removing the missing age cohorts and simulating fewer years or using extrapolation techniques for the missing data.
10. IQR intervals throughout manuscript: IQR intervals use inconsistent format throughout the article. I might suggest using a comma or semicolon to separate the percentiles in order to avoid confusion between an en dash and minus sign on negative values. In particular, the IQR in line 228 doesn't make sense to me as written and the IQR in line 245 is confusing since an increase in cases is being presented as a negative value.
11. Lines 296-298: This sentence initially confused me since I thought it was referring to model results until I got to "at the time" (which is also somewhat vague language) – I might suggest a rephrase that clearly sets off the beginning of the sentence to differentiate from the previous paragraph.
12. Lines 327-328: To a lay reader, an increase in the number of compartments is not clear as to why its infeasible – please clarify.
13. Please review the manuscript in detail for typographical and grammatical errors. For example, line 33 has a comma where there should be a period and line 243 says "adverted" where it should say "averted."

(Remarks on code availability)

Reviewer #2

(Remarks to the Author)

This study used mathematical modelling to understand the impact of the upcoming policy change to MMR vaccination in England, which will bring forward the age at second MMR dose (MMR2). The authors explore various scenarios to determine the impact of moving the timing of MMR2 forward or back, and raising coverage with the first MMR dose (MMR1) or MMR2 to various degrees. This work is a useful exercise in understanding the impact of policy changes using real world data.

Despite this being a thorough and well-executed study, there are several aspects of the mathematical model that limit our ability to use it for predicting what we may see in England upon this policy change. I provide some specific feedback below. I am also including some comments related to subject matter accuracy and terminology.

Introduction:

- Lines 44-45: The authors refer to eradication of measles in large geographical areas, but I think the authors are referring to elimination. These are two distinct technical terms. Elimination is not the same as geographic eradication, because the former allows endemic transmission for <12 months, while the latter means there is no measles present whatsoever, and is usually not limited by geography.

- Lines 45-46: The first measles vaccine was produced in 1963, not 1968.

- It would be more appropriate to cite a range for vaccine effectiveness of MMR1. There is no reference provided but 95% is likely much higher than most effectiveness estimates in the field.

- Line 62: I am familiar with Dr. Funk's study and agree it should certainly be cited, however, in the context of other work that was done. The way the sentence is written makes it seem like the concept of 95% coverage (for an entire population, not for a specific age-group) was not pre-existing.

Methods:

- One thing that is unclear from the methods is whether the model was built for the entirety of England (ie – using national case numbers and vaccine coverage) or using regional data. As the authors know, there were several regions in England (ie – West Midlands) that saw large outbreaks while in other areas there was much less activity, and regional coverage is often much more indicative of outbreak risk than national coverage estimates. The authors address this as a limitation in the Discussion, and I understand why they could not work regionality into their model, but it would be useful to be more explicit in the methods that no regional data were included. I agree that this is in fact a big limitation of the study that was not in the authors' control.

- Line 189: I understand the limitations of the age stratification in the model and the need to use 24 months of age for MMR2 in the simulations. However, it would be informative to understand whether there is a scheduled healthcare visit for English children at age 2 years (as in other countries, when children are invited for Well Child Visits around their birthday to ensure they are meeting milestones). This is important because the 18-month time point does NOT have a corresponding visit – so there is a need to know whether 24 months can be used as an appropriate comparison.

- Line 207: more information needs to be added on the waning immunity sensitivity analysis. All vaccines wane (ie, immunity markers decrease). Do the authors mean a reduction of antibodies to the point that individuals are seronegative? What timeframe was used, and which rate of seroreversion?

Results

- The results are very clearly written and the figures+figure legends illustrate the results very nicely. I think some of the later sections in the Results are a bit repetitive and the findings can be combined into fewer sections.

- Line 262: Do the authors perhaps mean figure 5, not figure 4? Figure 4 only goes up to C. Apologies if I am mistaken.

- I was surprised to see that bringing MMR2 forward was as effective at decreasing cases as raising MMR1 coverage by 0.25-0.5%. One would expect that, since the vast majority (90%+) of MMR1 recipients are already protected before receiving MMR2, the increase in population immunity would be marginal. I think this should be discussed in the discussion. Were the authors surprised by this finding? Do we think it correctly mimics what would happen in real life? If so, why? I wonder if it is related to contact patterns in young children.

- Waning is mentioned in several figure legends. The readers need to know more about what data were used to parameterize the rate of waning, which as the authors know depends on several factors.

Discussion

- Lines 288-295: I think that the rationale for the JCVI statement and the math model are a bit at odds: JCVI made the recommendation to move MMR2 forward to raise coverage of MMR2 (this may also result in a bump in MMR1 because it is another opportunity to discuss measles vaccine). The model does not address this, unfortunately. In other words, the model is not asking "how much higher would coverage be if we moved MMR2 up?". Rather, if I understand correctly, it is trying to simulate the number of cases resulting from this change (and other change scenarios). Therefore, it is not directly testing JCVI's hypothesis because the increases in coverage that are used in the simulation are arbitrary. Since the model is still very useful, I wonder if a more appropriate way to frame the findings would be "JCVI's recommendation will lower measles cases by 16% even if coverage is not increased as a result of the schedule change". If I am understanding the model correctly, that is a really powerful and impactful finding.

- Lines 296 - 308: I think that comparing a change in schedule to catch up campaigns is not appropriate. It is very likely that the catch-up campaigns focused on exposed groups, who would be more motivated to be vaccinated, and be biased in terms of who they are targeting compared to the general population.

- Lines 337-338: Whether or not population immunity wanes depends on several factors (as noted above) and is only possible if measles is not endemic, and may take decades. It would be helpful for the authors to expand on this point by relaying some of the topline findings in reference 20 in order to take into account this nuance.

- The authors should add some information on the Discussion about whether there are differences in cases in jurisdictions that implement MMR2 at different ages, but are otherwise similar (for example, some of the Nordic countries, or different jurisdictions in Canada or Mexico).

(Remarks on code availability)

The code looked ok to me but I did not go through it line by line, nor did I try to install and run in R.

Reviewer #3

(Remarks to the Author)

** Comments to the authors **

How is the model calibrated? The text says that details about the model fitting process can be found in the supplementary material (I150) but S5 does not appear to contain any information related to the model fitting process.

- What data is being used for the fit? Is it just yearly case data by region?

- What parameters are being fit? What are the fitted parameter values and do you have uncertainty estimates for them?

Does the model include any age mixing? If so, how is this informed or constrained?

Does the model include seasonality? If so, how is this informed or constrained?

Does the model include importations. If so, how is this informed or constrained?

The results are presented as mean and IQ values of cases from 250 evaluations of the model. Is the spread in model trajectories driven by the seed or do they also represent uncertainty from any of the model parameters (152-153)?

The text says that scenarios are matched using "the same seed for the stochastic simulations". However, in S5 the model is described as "deterministic". These appear to be incompatible descriptions. Can the authors clarify?

Since the analysis is focused on age at vaccination policy it is critical that the model is reproducing the age distribution of cases. Can the authors include model fits to the UK HAS data in addition to Figure 5?

Are improvements to cases similar across the different regions? (i.e., does London see the same relative gains in case reductions as regions with higher coverage like West Midlands?)

The majority of cases arise from 2011-2013. Given that the model is fixed at 2010 (all trajectories for all scenarios start at the same value) is it possible that the effect of the policies is underestimated? One way to explore this would be to evaluate the effect for a fixed year. Is the effect on measles consistent year by year?

Figure 1 shows a marked disjoint in coverage for both MMR1 and MMR2 between 2010 and 2011 for most regions. Is this due to the COVER data or can it be traced to a historical effect (e.g., change in policy, historical event, etc.). Is it possible to repeat the analysis only using the CPRD data? And then the COVER data? It also looks like it would be possible to estimate the region specific underascertainment of COVER based on the comparison with the CPRD data. Is there a reason not to do this and to prefer the 50% adjustment (from Suffel et al. 2023)?

It is unclear from the text and figures whether the increase in MMR1 and MMR2 simulated in the scenarios is a relative or absolute gain and whether it is year-to-year or over the course of the simulation. Some regions see a decline in coverage from 2010-2019 so do these scenarios use the full historical trajectory and simply amend each year or do you fix 2010 and then increase from that starting point? Please clarify. It might also be helpful to overplot the trajectories in Figure 1. Consider updating Figure 1 by (1) moving the year to the x-axis (improved clarity and black and white printing), (2) only including the 8 regions considered (simplifies the figure and avoids confusion with 10 subplots since the simulation is

stratified by region), and (3) use the same y axis scale for MMR1 and MMR2.

The title of the talk suggests that the results here are relevant to low-incidence settings. However the analysis is purely focused on the UK setting without substantive discussion or work on how it might extend to other locations. It seems more appropriate to set the context in the title "How vaccine timing, timeliness, and coverage impact measles outbreaks in low incidence settings".

Further along this point, the analysis does not go into the timeliness (e.g., children getting their vaccine at a time different than the policy) and only the timing (e.g., what is the policy). It is confusing to have "timeliness" in the title.

Lastly, the results look purely at number of cases and don't have much to say about the nature (e.g., number or severity) of outbreaks. Having the word "outbreak" in the title is also somewhat confusing.

Why is the github repository "measles_england_sir" when the model is an SEIR model? The repository is also a branch on the repository for Robert et al. 2024. I'd recommend making a separate repository to avoid confusion.

For their analysis, Roberts et al. used 5k samples per scenario. Why does this analysis only use 250?

(Remarks on code availability)

The paper lists the project at: https://github.com/alxsrobert/measles_england_sir/tree/vaccination_scenarios/R

The code exists but is not well documented. Potential improvements include:

- 1) A separate repository rather than a branch
- 2) Adding a README.md in the R/ folder (if that is the project folder) detailing the steps to reproduce the analysis (or at least explaining what each of the scripts is meant to do and a rough ordering of the code)
- 3) A LICENSE

Version 1:

Reviewer comments:

Reviewer #1

(Remarks to the Author)

The authors have adequately responded to all of my previous comments.

(Remarks on code availability)

Reviewer #3

(Remarks to the Author)

See attached

(Remarks on code availability)

The new repository was an improvement. With some edits I was able to get the code to run and reproduce the figures. The paper claims that they are inhibited by computation but, from what I can tell, the code is not well optimized.

Version 2:

Reviewer comments:

Reviewer #3

(Remarks to the Author)

Thank you for your responses and engagement during the review process. I have no additional questions.

(Remarks on code availability)

We thank all of the reviewers for their detailed comments and incredibly helpful suggestions. We have responded to each comment point by point in the text below. As we needed to rerun the simulations in response to the reviewers' comments, the figures have been updated accordingly.

REVIEWER COMMENTS

Reviewer #1 (Remarks to the Author):

I read this paper with great interest as a valuable addition to the literature. However, I believe that revisions are needed prior to publication, as described in the following comments.

1. Lines 44-45: This sentence is not technically correct as written, since eradication has not been achieved. Suggest a rephrase such as, "The global eradication of measles through vaccination is feasible and elimination has been achieved in large geographical areas."

We agree with the reviewer and rephrased the sentence to the following:

"The global eradication of measles through vaccination is feasible and several countries have already achieved measles elimination (2)."

2. Lines 47-48: The sentence specifying 95% vaccine effectiveness is missing a citation. A citation for this sentence is essential.

We have added the citations accordingly.

3. Lines 48-50: I think it is somewhat misleading to say that MMR1 should be administered around the first birthday, since this is really the recommendation regarding mumps whereas this paper is focused on measles. The WHO recommendation is 9 or 12 months, minimum 6 months, for measles. More context and clarification is needed here.

We clarified that this applies to countries with lower incidence of measles, such as the UK and other European countries.

"The first dose of vaccine (MMR1) should be administered around the first birthday in countries with low incidence (7), as earlier first dose vaccination can reduce the vaccine effectiveness"

4. Lines 123-132: I am not sure that I fully follow the logic here. If COVER data is not fully representative and not quality-assured, and you are correcting

values based on CPRD data, wouldn't a scenario in which you extrapolate the CPRD data for the missing cohorts be valuable? It might be that this section needs further elucidation.

CPRD data is based on the EMIS © GP software which was not widely used before 2006 and only available for children born up to 2015. Due to recent publications by UKHSA about the underascertainment of COVER data (UK Health Security Agency, 2023), we consider CPRD the more accurate data source for vaccine coverage.

Hence, we started to use CPRD data from the first year where sufficient data coverage was available on CPRD, i.e., 2006. In order to include data on vaccination coverage for other age bands, we needed to supplement missing CPRD data with estimates from the COVER data which is available for more birth cohorts. As COVER is not fully quality assured, we followed recent publications by UK Health Security Agency which suggested to correct the COVER data for 50% underascertainment.

We followed the reviewer's suggestion and clarified this in the Vaccine Data section.

“For children born before 2006 or after 2015, the CPRD vaccine data had to be supplemented with estimated values from COVER data. COVER uses on aggregated GP information based on operational data which may be incomplete, not fully representative and not quality assured (32). COVER data that has been corrected for underascertainment is closer to CPRD estimates. A comparison between COVER data and CPRD data can be found in the supplementary material (see Supplementary Section S2). Based on this, COVER estimates used to supplement missing CPRD data were adjusted using the assumption that 50% of the unvaccinated children were vaccinated but did not have their vaccine recorded (15). These corrected values were consistent with estimates from previous studies (33). To estimate the vaccine coverage in the missing age bands in the COVER data, we applied the relative difference of the age bands for the last completely available years, i.e., 2006 and 2017, to the COVER estimates to supplement the values for ages three and four. All these values were stratified by region (see Figure 1).

In a sensitivity analysis, we fitted the model to the uncorrected COVER data for which the missing age strata were supplemented by the proportional change in uptake between age groups as observed in the CPRD data.”

“

5. Reference #20: I went to check this paper referenced for the original transmission model, and it seems like the citation is incomplete in the bibliography.

We corrected this reference.

6. Lines 150-152: Supplementary exhibit S5 does not clearly define what parameters were varied and under what assumptions. If 250 simulations per vaccination per vaccination strategy were run, it is important to include in this paper what elements could vary between each simulation.

We ran 2500 simulations which is now corrected in the manuscript. Further, we expanded the paragraph on which parameters were stochastically derived for each simulations, and hence, might vary across the different simulations.

“We ran 2500 simulations per vaccination strategy by drawing 100 parameter sets from the model fits, and running 25 simulations per parameter set. Parameters drawn from the model fits included infection rate, duration of maternal immunity, parameters for seasonality of transmission and importation, report rate of imported cases, vaccine effectiveness, existing immunity in older generations and parameters of spatial spread (see Table S7). In the stochastic simulations, the number of transitions was computed using binomial draws, with a rate of transition derived from the distribution of the population between compartments at the previous time step, and the parameter set. Counterfactual scenarios were matched by using the same seed for the stochastic simulations. “

7. Lines 187-189: If 1-year age bands are used and the model assumed 24 months rather than 18 months for the alternative scenario, how was the model able to account for a schedule of 3 years 4 months in the reference scenario? This needs to be clarified.

We added a paragraph to the description of the vaccination data to explain how we dealt with the age-band structure of the model considering the vaccination schedule not corresponding to the same age bands.

“As the recommended age for MMR vaccination is one year (MMR 1) and three years and four months (MMR 2), most children receive a first dose between one and two, and a second dose between three and four. Therefore, the vaccination coverage for MMR1 at age 1 and MMR2 at age 3 is nearly zero in the data, but a large proportion of children aged one have actually received a dose of vaccine (same with MMR2 for children aged three). To adjust the coverage data in the 1-year age band structure of the model, we assumed the coverage of MMR1 at the age of 1 was 75% of the coverage for MM1 at age 2, and 50% of the coverage at age 3 of the coverage of MMR2 at the age 4.”

8. Section 2.5, Lines 160-202: In general, this entire section was quite difficult to follow. I might suggest revising Table 1 to more clearly delineate the different sets of assumptions for each scenario (e.g., a column for coverage, a column for MMR2 schedule), so that the text can more clearly link to the table and provide the additional information needed. The text in this section needs more clarity.

We followed the suggestion of the reviewer and updated Table 1 one which now included separate columns for the schedule of MMR2, the coverage of MMR1 and the coverage of MMR2 in each scenario. We also changed the first paragraph of the section to clarify this section:

“We explored how changes in vaccination impacted the number of cases simulated by the transmission model. To do so, we generated sets of stochastic simulations using the same parameter sets as the reference simulation set (described in section 2.4), but with changes in vaccination schedule, vaccine coverage, or both. Changes in schedule were based on other vaccination schedules implemented in Europe.”

9. Lines 205-208: This comment connects to comment #4 above. If sensitivity analyses with alternative assumptions are included, I might suggest an additional scenario that only uses CPRD data – either removing the missing age cohorts and simulating fewer years or using extrapolation techniques for the missing data.

For the transmission model it is necessary to consider the full immunity landscape in the population. . Neither CPRD nor COVER offer a full description of historical vaccine coverage in the population. We must use a combination of COVER and CPRD data to best estimate the level of coverage in each age group:

- 1) CPRD is thought to reflect the vaccine uptake in the population more accurately as it followed up individuals from birth record to fifth birthday with better assessment of the underlying population denominators than the COVER data (Suffel et al., 2024; UK Health Security Agency, 2023). However, this data was only available for children born between 2010 and 2015 at the study. Unfortunately this age range is insufficient to run the model even for 2015 alone, since the transmission model would also require the vaccination status of children older than 5 years to take the immunity landscape into account. Hence, the use of COVER data was necessary to supplement the vaccination coverage estimates for missing years and older generations
- 2) COVER data for the MMR vaccine is only collected for children aged two and five. As the model has one-year age-strata between the ages one to five, the use of CPRD data was necessary to estimate vaccination coverage for the

ages one, three and four, and to approximate of uptake in these age groups for the COVER data.

In order to reflect the strength and weaknesses of each dataset, we conducted the main analysis using CPRD data but supplemented with adjusted COVER data for missing age cohorts and conducted a sensitivity analysis using the COVER data but approximating missing age strata (i.e., age one, three and four) using CPRD data.

We changed the description of the vaccine data in order to clarify that we have to use both data sources:

“Two data sources of vaccine coverage were used: The Clinical Practice Research Datalink (CPRD) Aurum to estimate vaccine coverage by region and one-year age bands and the Cover of Vaccination Evaluated Rapidly (COVER) to supplement missing data for age groups not included in CPRD Aurum.”

10. IQR intervals throughout manuscript: IQR intervals use inconsistent format throughout the article. I might suggest using a comma or semicolon to separate the percentiles in order to avoid confusion between an en dash and minus sign on negative values. In particular, the IQR in line 228 doesn't make sense to me as written and the IQR in line 245 is confusing since an increase in cases is being presented as a negative value.

We agree with the reviewer and now present all IQR using semicolons only in order to avoid confusion with negative values.

11. Lines 296-298: This sentence initially confused me since I thought it was referring to model results until I got to “at the time” (which is also somewhat vague language) – I might suggest a rephrase that clearly sets off the beginning of the sentence to differentiate from the previous paragraph.

We rephased this sentence which now reads:

“A previous study has shown that the implementation of accelerated MMR immunization schedules in some boroughs of London (23) resulted in an increase of vaccine coverage although the overall lower coverage remained lower than in the rest of the country (31,33).”

12. Lines 327-328: To a lay reader, an increase in the number of compartments is not clear as to why its infeasible – please clarify.

We have explained now that an increased number of compartments would make the model fitting computationally not feasible anymore:

“Additional spatial granularity would multiply the number of strata in the model, leading to an increase of compartments which would greatly slow down model fitting, making fitting computationally unfeasible. “

13. Please review the manuscript in detail for typographical and grammatical errors. For example, line 33 has a comma where there should be a period and line 243 says “adverted” where it should say “averted.”

Thank you! We corrected these errors.

Reviewer #2 (Remarks to the Author):

This study used mathematical modelling to understand the impact of the upcoming policy change to MMR vaccination in England, which will bring forward the age at second MMR dose (MMR2). The authors explore various scenarios to determine the impact of moving the timing of MMR2 forward or back, and raising coverage with the first MMR dose (MMR1) or MMR2 to various degrees. This work is a useful exercise in understanding the impact of policy changes using real world data.

Despite this being a thorough and well-executed study, there are several aspects of the mathematical model that limit our ability to use it for predicting what we may see in England upon this policy change. I provide some specific feedback below. I am also including some comments related to subject matter accuracy and terminology.

Introduction:

- Lines 44-45: The authors refer to eradication of measles in large geographical areas, but I think the authors are referring to elimination. These are two distinct technical terms. Elimination is not the same as geographic eradication, because the former allows endemic transmission for <12 months, while the latter means there is no measles present whatsoever, and is usually not limited by geography.

We clarified the language of this sentence, as we are indeed referring to elimination. This sentence now reads: *“The global eradication of measles through vaccination is feasible and several countries have already achieved measles elimination (2).”*

- Lines 45-46: The first measles vaccine was produced in 1963, not 1968.

The reviewer is correct, we fixed this typo.

- It would be more appropriate to cite a range for vaccine effectiveness of MMR1. There is no reference provided but 95% is likely much higher than most effectiveness estimates in the field.

We based this information on an official publication by the European Centre of Disease Control using 95% and a systematic review and meta-analysis by Schenk et

al. which estimates the seroconversion rate for the measles component of the MMR vaccine at 96.0% (95% CI: 94.5-97.4). We have added the citations accordingly.

- Line 62: I am familiar with Dr. Funk’s study and agree it should certainly be cited, however, in the context of other work that was done. The way the sentence is written makes it seem like the concept of 95% coverage (for an entire population, not for a specific age-group) was not pre-existing.

We corrected the sentence, to emphasise it was the Funk et al paper which added the age-group aspect for gaining herd immunity through vaccination. The section now reads as the following:

“Since measles is highly infectious, high vaccine coverage is required to mitigate the risks of outbreaks – several studies estimated that a coverage of around 95% (12,13), achieved by the age of five, was necessary to ensure elimination of measles transmission (14).”

Methods:

- One thing that is unclear from the methods is whether the model was built for the entirety of England (ie – using national case numbers and vaccine coverage) or using regional data. As the authors know, there were several regions in England (ie – West Midlands) that saw large outbreaks while in other areas there was much less activity, and regional coverage is often much more indicative of outbreak risk than national coverage estimates. The authors address this as a limitation in the Discussion, and I understand why they could not work regionality into their model, but it would be useful to be more explicit in the methods that no regional data were included. I agree that this is in fact a big limitation of the study that was not in the authors’ control.

The model was stratified by the nine English regions, with case and vaccination data for each region. To clarify this in the paper, we more explicitly state the model strata in the study design section which now reads as:

“We generated stochastic simulations using a mechanistic transmission model that had previously been fitted to the daily number of confirmed cases reported in England stratified by age, region and vaccination status (26).”

We also modified the description of the vaccine data (2.3) to be more explicit of the use of regional data.

“Using a validated algorithm to identify vaccination records (29), vaccination coverage at the ages of 1, 2, 3, 4, and 5 years stratified by region was estimated from the electronic health records. “

“Two data sources of vaccine coverage were used: The Clinical Practice Research Datalink (CPRD) Aurum to estimate vaccine coverage by region and one-year age bands and the Cover of Vaccination Evaluated Rapidly (COVER) to supplement missing data for age groups not included in CPRD Aurum.”

However, as highlighted in the Discussion, we believe that the limitation on the lack of spatial granularity still applies: heterogeneity in vaccine coverage within regions cannot be captured by the model, and is likely to drive local outbreaks in pockets of susceptibility.

“There are several limitations to this approach. Firstly, although the model is stratified by region, it lacks spatial granularity: Outbreaks in low incidence settings are usually driven by heterogeneity in vaccination in certain groups and communities that cannot be captured on a regional scale (41,42).”

- Line 189: I understand the limitations of the age stratification in the model and the need to use 24 months of age for MMR2 in the simulations. However, it would be informative to understand whether there is a scheduled healthcare visit for English children at age 2 years (as in other countries, when children are invited for Well Child Visits around their birthday to ensure they are meeting milestones). This is important because the 18-month time point does NOT have a corresponding visit – so there is a need to know whether 24 months can be used as an appropriate comparison.

There are five health visits scheduled for children in England up to the age of 2 years and 6 months. The fourth health visit is usually scheduled between 9 and 15 months of age and the last visit is scheduled for a child aged 24 to 30 months. Only 56.8% children in London receive a fourth visit and only 44% a fifth visit, whereas the number of visits is significantly higher in other English regions (https://assets.publishing.service.gov.uk/media/5a75ab1440f0b67f59fcae7/Review_of_mandation_universal_health_visiting_service.pdf).

We agree with the reviewer that a vaccination appointment scheduled together with a healthcare visit might result in higher vaccination uptake. Hence, a vaccination recommended around the same time as the last health visit might result in slightly better uptake than a vaccine recommended 3 months after the last health visit. For this reason, we tested several scenarios with varying vaccine coverage while bringing the second MMR dose forward.

We added a sentence to the discussion, discussion how scheduling the second MMR vaccine at the same time as a health visit might be influence uptake :

“Determinants of childhood vaccine uptake are complex, and include parental vaccine confidence, opportunities to ask questions, reminders and barriers to vaccination appointments, which could all be impacted by changes in vaccine schedules (35). However, vaccine uptake is generally higher for the earlier appointments in the UK childhood vaccination schedule and so it may be reasonable to assume that an earlier MMR2 date is likely to result in higher uptake (33). Introducing an earlier second MMR2 at a similar time to a routine health visit for the child might also offer additional opportunity to remind parents of the vaccination and provide information.”

- Line 207: more information needs to be added on the waning immunity sensitivity analysis. All vaccines wane (ie, immunity markers decrease). Do the authors mean a reduction of antibodies to the point that individuals are seronegative? What timeframe was used, and which rate of seroreversion?

Thank you for this comment. The waning explored in the sensitivity analysis refers to a waning of protection against infection. In absence of waning, individuals who develop protection upon vaccination (i.e. excluding primary vaccine failures) have full, lifelong protection. In scenarios with waning, individuals who develop protection after vaccination have a risk of becoming infected, with the risk increasing with age. Age then acts as a proxy for time since vaccination. The mechanism of waning in the model is explained in Supplementary section S3.

This waning scenario was based on our previous work (Robert et al., 2024b) which estimated an annual waning rate of approximately 0.03% starting from the age of five. This also aligns with existing literature on waning of the measles vaccine (Guerra et al., 2018; Yang et al., 2020). If waning starts at the age of five, this means that the vaccine effectiveness declines every year by 0.03% for an individuals which initially responded to the vaccine (i.e., out of 1000 double vaccinated individuals aged 25 with no primary vaccine failure exposed to measles, on average 6 individuals would get infected). To take the earlier administration of MMR2 into consideration of the context of waning immunity, we added a second sensitivity analysis which included waning from the age of three years.

We clarified this in the methods.

“2) We included waning of vaccine-induced immunity from age 5, whereby vaccinated individuals who developed immunity upon vaccination can become infected, with infection risk increasing with age. In the model with waning, age is used as a proxy for time since vaccination, and the waning rate based on estimates of our previous work (26); 3) as moving MMR2 delivery to age 2 may influence when waning begins, we

ran a sensitivity analysis with waning of immunity starting at age 3 when MMR2 was given early (section S6 in the supplement). (26)”

Results

- The results are very clearly written and the figures+figure legends illustrate the results very nicely. I think some of the later sections in the Results are a bit repetitive and the findings can be combined into fewer sections.

We thank the reviewer for this comment. We removed the subsection on “comparing timing and coverage” to avoid repeating findings.

- Line 262: Do the authors perhaps mean figure 5, not figure 4? Figure 4 only goes up to C. Apologies if I am mistaken.

This is correct, this should refer to Figure 5. We corrected this typo.

- I was surprised to see that bringing MMR2 forward was as effective at decreasing cases as raising MMR1 coverage by 0.25-0.5%. One would expect that, since the vast majority (90%+) of MMR1 recipients are already protected before receiving MMR2, the increase in population immunity would be marginal. I think this should be discussed in the discussion. Were the authors surprised by this finding? Do we think it correctly mimics what would happen in real life? If so, why? I wonder if it is related to contact patterns in young children.

We explained this point in more detail in our discussion:

“In the reference scenario without waning, the risk of primary vaccine failure estimated when fitting the model to the case data is 5.2% (95% Credible Interval (CI): 4.9%; 5.5%). In the early MMR2 scenario with no change in coverage, most vaccinated children aged 2 and 3 have received 2 doses of vaccine, leading to a reduction of vaccinated children with primary vaccine failure. This increased immunity in children aged 2 and 3 causes indirect protection to the rest of the population, leading to an overall 14.5% reduction of cases compared to the median number of cases in the reference simulations (IQR: -0.01%-26.03%). When waning is included in the model, the risk of primary vaccine failure estimated by the model is lower (2.5%, 95% CI 2.2% ; 2.9%), and the impact of early MMR2 on overall case number is also slightly lower (12.25% (-2.26; 24.64)). It is difficult to assess which model is closer to the true impact of early MMR2 as both reference simulation sets slightly diverge from the data: the number of measles cases aged 2 and 3 was underestimated in the model with waning (315 (IQR: 266-373) cases, while 486 were observed in the data), hence it might

underestimate the protective effect of bringing the second dose forward. The model without waning overestimated the number of vaccinated cases aged 2 and 3 (164 (IQR: 140; 193) cases, while 52 were observed in the data), hence it might overestimate the protective effect of an earlier second dose. We conclude that the number of cases in early MMR2 vaccination scenario was consistently 10% lower than the median number of cases in the reference simulations, and that the decrease was robust to changes in MMR2 coverage.”

- Waning is mentioned in several figure legends. The readers need to know more about what data were used to parameterize the rate of waning, which as the authors know depends on several factors.

The legend of Figure 5 should actually not contain references to waning, this has been modified. Only figures and tables describing the sensitivity analysis including waning mention waning (Supplementary section S5).

The rate of waning was estimated by fitting the mathematical model to English measles case data between 2010 and 2019 (Robert et al., 2024a). We have added more information on this under the description of the methods section (see previous comment) and in the Supplement (Supplementary section S3). Waning of immunity is influenced by various factors such as age at vaccination and circulation of the wild-type measles virus (Hughes et al., 2020), which was assumed to be homogeneous in the population.

Discussion

- Lines 288-295: I think that the rationale for the JCVI statement and the math model are a bit at odds: JCVI made the recommendation to move MMR2 forward to raise coverage of MMR2 (this may also result in a bump in MMR1 because it is another opportunity to discuss measles vaccine). The model does not address this, unfortunately. In other words, the model is not asking “how much higher would coverage be if we moved MMR2 up?”. Rather, if I understand correctly, it is trying to simulate the number of cases resulting from this change (and other change scenarios). Therefore, it is not directly testing JCVI’s hypothesis because the increases in coverage that are used in the simulation are arbitrary. Since the model is still very useful, I wonder if a more appropriate way to frame the findings would be “JCVI’s recommendation will lower measles cases by 16% even if coverage is not increased as a result of the schedule change”. If I am understanding the model correctly, that is a really powerful and impactful finding.

We agree with the reviewer that we should be clearer with our hypothesis in relation to the JCVI statement. We rephrased the sentence of the discussion accordingly:

“We explored how different vaccination schedules could impact the risk and dynamics of measles outbreaks and showed that advancing the second dose of MMR would still result in a reduced number of measles cases even if the uptake did not improve as hoped by the policy recommendation from JCVI (22). “

- Lines 296 - 308: I think that comparing a change in schedule to catch up campaigns is not appropriate. It is very likely that the catch-up campaigns focused on exposed groups, who would be more motivated to be vaccinated, and be biased in terms of who they are targeting compared to the general population.

We rephrased this sentence to focus only on the accelerated schedules as implemented in some boroughs of London which are not targeted to specific exposed groups.

The paragraph now reads as:

“A previous study has shown that the implementation of accelerated MMR immunization schedules in some boroughs of London (23) resulted in an increase of vaccine coverage although the overall coverage remained lower than in the rest of the country (31,33). However, this only works under the assumption that an additional vaccination appointment will be accepted by parents as the MMR2 is currently given at the same time as the 4-in-1 preschool booster. Determinants of childhood vaccine uptake are complex, and include parental vaccine confidence, opportunities to ask questions, reminders and barriers to vaccination appointments, which could all be impacted by changes in vaccine schedules (35). However, vaccine uptake is generally higher for the earlier appointments in the UK childhood vaccination schedule and so it may be reasonable to assume that an earlier MMR2 date is likely to result in higher uptake (33). Introducing an earlier second MMR2 at a similar time of a health visit for the child might also offer additional opportunity to remind parents of the vaccination and provide information.”

- Lines 337-338: Whether or not population immunity wanes depends on several factors (as noted above) and is only possible if measles is not endemic, and may take decades. It would be helpful for the authors to expand on this point by relaying some of the topline findings in reference 20 in order to take into account this nuance.

We clarified this point and added a sentence that this only applied to non-endemic settings.

“A second vaccine dose given at younger age might lead to marginally less protection in adults in the long term in a non-endemic setting (26,42). In a sensitivity analysis, we showed that the median number of cases with early MMR2 vaccination remained 9% smaller than the reference simulations when waning started at 3 years instead of 5. This shows that the short and medium term consequences of waning do not outweigh the impact of early MMR2 vaccination in our model. However, close monitoring of transmission in vaccinated adults must be maintained to understand long-term immunity of vaccinated individuals in countries near elimination.”

- The authors should add some information on the Discussion about whether there are differences in cases in jurisdictions that implement MMR2 at different ages, but are otherwise similar (for example, some of the Nordic countries, or different jurisdictions in Canada or Mexico).

There is no obvious pattern for the rise of measles cases in Europe for comparable countries but different vaccination schedules (e.g., Sweden, Norway, Finland and Denmark), but comparing countries is not straightforward: incidence, historical vaccine coverage, spatial distribution of coverage, age structure of the population can all influence transmission and confound direct comparisons. Further work is needed to understand how vaccine schedule explains measles incidence.

We added the following sentence to the discussion of the paper:

“More work is needed to understand how vaccination schedules influences outbreak risks in near elimination countries: Vaccination schedules differ in near elimination countries, but differences in historical coverage, previous incidence, age structure of the population, and spatial distribution of coverage make direct comparison challenging. With the increase in measles cases observed across most European countries in 2023 and 2024, it is important to understand whether early MMR2 schedule is consistently associated with lower outbreak risk (39).”

Reviewer #2 (Remarks on code availability):

The code looked ok to me but I did not go through it line by line, nor did I try to install and run in R.

Following the comments of reviewer #3, we created a separate repo with more detailed information on how to run the code and all the simulations generated which allow the reviewers to check our results and recreate figures and tables (https://github.com/Eyedeet/measles_vaccination_scenarios).

REVIEWER 3 full report:

**** Summary ****

This paper assesses the impact of various policies for the recommended age of measles vaccination in England (specifically the second dose, MMR2). They utilize a compartmental model of measles transmission fit to case data from 2010-2019 stratified by the 8 regions in England. The authors conduct counterfactual analyses assessing how cases would have changed for a suite of vaccination coverage and vaccination age policies for MMR2. The analysis is relevant given the recent JCVI recommendation to bring MMR2 forward to 18 months by 2025.

The paper is promising and well-constructed. However, it is missing some critical information regarding the model without which it is difficult to assess its validity and relevance to the question at hand.

**** Strengths: ****

The analysis is timely given the JCVI recommendation to bring MMR2 forward to 18 months. The JCVI recommendation was motivated by evidence that this might increase coverage; this analysis also suggests that earlier timing (18 month MMR2 recommendation) may also reduce overall cases.

The results include supporting studies including waning and school entry scenarios.

The authors utilize two sources to estimate vaccination coverage. Most of the time span is informed by electronic health records (CPRD). However, for times when the records are not available the authors utilize summaries from NHS Digital (COVER).

Weaknesses:

The text is missing some information regarding calibration and model relevance. The model appears to be lifted from another recent study by the same authors (Roberts et al. 2024). However, the paper under consideration does not provide the details necessary to assess the model (and consequently the analysis) directly from the text. In my opinion it is not sufficient to reference another paper for essential model details such as parameters and calibration methodology. When I look at the paper by Roberts et al. 2024 it is not entirely clear which of the fitted models this analysis is using. Please see my comments to the authors.

We have added more information on the calibration and fitting of the model in the Supplement and the Main Paper, along with figures comparing the simulations and the case data. See specific answers to the comments made by the reviewer,

Some of the figures could be improved for clarity.

The figures were updated according to specific points made by the reviewers.

The analysis is purely counterfactual, focusing on recent historical cases. The study cannot comment on how future projections of demographic, vaccination trends, or case importation may impact potential measles cases with the considered policies.

This point has been added to the limitations of the paper *“Finally, this analysis focuses on counterfactual scenarios of measles transmission between 2010 and 2019, and does not account for the impact of vaccination trend, demographics, or global measles dynamics on future outbreaks in England”*.

It is hard to follow how yearly coverage at age 5 (e.g., figure 1) maps onto vaccination rates for the different age bins. This is particularly important since the authors are using 2 different data sources for their coverage data (CPRD and COVER).

We updated figure 1 to better illustrate the CPRD vaccine data by English region over time and we clarified the mapping of the coverage data to the age bins in the methods section, which now reads as:

“As the recommended age for MMR vaccination is one year (MMR 1) and three years and four months (MMR 2), most children receive a first dose between one and two, and a second dose between three and four. Therefore, the vaccination coverage for MMR1 at age 1 and MMR2 at age 3 is nearly zero in the data, but a large proportion of children aged one have actually received a dose of vaccine (same with MMR2 for children aged three). To adjust the coverage data in the 1-year age band structure of the model, we assumed the coverage of MMR1 at the age of 1 was 75% of the coverage for MM1 at age 2, and 50% of the coverage at age 3 of the coverage of MMR2 at the age 4.”

The study is limited to England and it is unclear how much is relevant to low incidence settings. The title in general seems to be somewhat misaligned with the focus of the paper. Please see my comments to the authors.

We changed the title of the paper according to the reviewer’s comment (see below) and added some more information on how these findings are more generalisable to other low-incidence settings (see response to comment below)

The study only briefly comments on the tradeoff of boosting effect w/ age (1337-338) and does not consider at all the potential reduction in outbreaks in school/pre-school settings. Without discussion of this the paper feels incomplete particularly w/ the evidence of waning from Roberts et al. (2024). Thinking more long term, the potential effect of boosting strength is potentially critical.

We have added an extra sensitivity analysis exploring the short and medium-term impact of early waning as a consequence of the change in vaccination schedule, and discussed its impact. However, this analysis relies on the assumption that the waning rate is not affected by the age of vaccination (i.e. waning starts early but is not faster), if changing the vaccination schedule impacted the waning rate, then the tradeoffs may be more balanced:

“We showed that the median number of cases with early MMR2 vaccination remained 9% smaller than the reference simulations when waning started at 3 years instead of 5. This shows that the short and medium term consequences of waning do not outweigh the impact of early MMR2 vaccination in our model. However, close monitoring of transmission in vaccinated adults must be maintained to understand long-term immunity of vaccinated individuals in countries near elimination.”

**** Comments to the authors ****

How is the model calibrated? The text says that details about the model fitting process can be found in the supplementary material (I150) but S5 does not appear to contain any information related to the model fitting process.

- What data is being used for the fit? Is it just yearly case data by region?

- What parameters are being fit? What are the fitted parameter values and do you have uncertainty estimates for them?

We added more detail on the model to the study design, clarifying that it has been fitted to daily cases by region, age and vaccination status.

“We generated stochastic simulations using a mechanistic transmission model that had previously been fitted to the daily number of confirmed cases reported in England stratified by age and region with cases by vaccination status (26).”

We further added more explanation to the description of the transmission model in the main text how the stochastic simulations were generated:

“A more detailed description of the model fitting process and model parameters can be found in the supplementary material (see S3). We ran 2500 simulations per vaccination strategy by drawing 100 parameter sets from the model fits and 25

simulations per parameter set. Parameters drawn from the model fits included infection rate, duration of maternal immunity, parameters for seasonality and importation, report rate of imported cases, vaccine effectiveness, existing immunity in older generations and parameters of spatial spread (see Table S7). In the stochastic simulations, the number of transitions was computed using binomial draws, with a rate of transition derived from the distribution of the population between compartments at the previous time step, and the parameter set. Counterfactual scenarios were matched by using the same seed for the stochastic simulations. “

We added more details to Supplementary Section S3, to explain how age and spatial contact rates are estimated in the model, we also , listed the parameters of the model and presented the model estimates (see tables S2 and S3).

Does the model include any age mixing? If so, how is this informed or constrained?

Does the model include seasonality? If so, how is this informed or constrained?

Does the model include importations. If so, how is this informed or constrained?

The model included age mixing, seasonality parameters and importations. The age mixing was informed by the age-stratified contact matrix from the POLYMOD study. Within-year seasonality of transmission was informed by two parameters estimated by the model.

Importations are also included in the model: The average yearly number of importations per region was computed from the number of cases classified as “imported” or “import related” in the individual case data for each region and year. This local number of importations for each year is then divided by 365 and weighted by the number of inhabitants per age group, to get the daily importation rate by age group and region. We considered that importations are less likely to be reported than other cases, so the importation rate by region is divided by p_{import} , the probability of reporting of imported cases. The model also estimates two parameters X_{import} and Y_{import} , to estimate the within-year seasonality of importations. The model therefore estimates three parameters to compute the number of importations: p_{import} , X_{import} and Y_{import} .

We added this information in the supplementary material (S3) which reads as:

“Contacts between age groups were included to the model based on the contact matrix from the POLYMOD study (3).

Contacts across the nine regions of England were approximated using a spatial kernel functions (see estimated parameters in table S7). This kernel is a gravity model, depending on population in both regions, and distance. Distance is

accounted for by the degree of connectivity between regions (neighbours have a degree of one, neighbours of neighbours have a degree of two etc.).

The within-year seasonality of transmission was estimated by two parameters (X and Y). For each time t , the infection rate (β) is computed as $\beta_t = \beta * (1 + X * \cos(\frac{2 * \pi * t}{365.25} + Y))$.

The average number of importations per year and region was computed using the number of cases who were classified as “imported” or “import related” in the individual case data. The daily importation rate by age group and region was then computed by dividing the local number of importations for each year by 365 and weighted by the number of inhabitants per age group. As importations may be less likely to be reported than other cases, the importation rate by region is divided by p_import , the report rate of imported cases. The model also estimates two parameters X_import and Y_import , to estimate the seasonality of importations in a given year.

$$n_{import}(a, i, t) = \frac{n_{import}(i, year(t))}{365 * p_{import}} * \frac{N_{ai}}{N_i} * (1 + X_{import} * \cos(\frac{2 * \pi * t}{365.25} + Y_{import}))$$

With N_{ai} the number of inhabitants of age a in i , N_i the number of inhabitants in region i , $n_{import}(i, year(t))$ the number of cases classified as “imported” or “import related” in the individual case data at each year. (Mossong et al., 2008) (Mossong et al., 2008)

We further describe the estimated parameters in the new supplementary tables S2 and S3, and figure S7 which describes importations and seasonality in more detail.

The results are presented as mean and IQ values of cases from 250 evaluations of the model. Is the spread in model trajectories driven by the seed or do they also represent uncertainty from any of the model parameters (152-153)?

We drew 100 parameter sets from the model estimates, and generated 25 simulations for each set, which amounts to 2,500 simulations in total. At each time step and for each compartment, the number of transitions between compartments is drawn from a binomial distribution, with the rate computed from the distribution of the population between compartments, and the parameter set. The spread in model trajectories comes from 1/ differences between the parameter sets, and 2/ stochasticity in binomial draws.

We have expanded on this in both the main text and the supplementary material, proving a clearer description which parameters are estimated and where the uncertainty of the simulations comes from.

“We ran 2500 simulations per vaccination strategy by drawing 100 parameter sets from the model fits and running 25 simulations per parameter set. Parameters drawn from the model fits included infection rate, duration of maternal immunity, parameters for seasonality of transmission and importation, report rate of imported cases, vaccine

effectiveness, existing immunity in older generations and parameters of spatial spread (see Table S7). In the stochastic simulations, the number of transitions was computed using a binomial draw

The text says that scenarios are matched using “the same seed for the stochastic simulations”. However, in S5 the model is described as “deterministic”. These appear to be incompatible descriptions. Can the authors clarify?

We fitted a deterministic compartmental model in the first place. However, the simulations are stochastic, using different particles and drawing different sets of parameters for the different simulations.

This is also now clarified in the methods:

“We generated stochastic simulations using a mechanistic transmission model that had previously been fitted to the daily number of confirmed cases reported in England stratified by age, region, and vaccination status (26). We used the parameter estimates obtained from the deterministic model fits to simulate stochastic outbreaks between 2010 and 2019.”

Since the analysis is focused on age at vaccination policy it is critical that the model is reproducing the age distribution of cases. Can the authors include model fits to the UK HAS data in addition to Figure 5?

Figure 3 of the manuscripts shows the comparison of the UKHSA case data (see dots) to the model fit of the reference scenario (CPRD data without waning). We agree with the reviewer that it is important to show the comparison between the stochastic simulations and the data, especially with regards to the age distribution of the cases, but also did not want to repeat too much of the content from our previous analysis (Robert et al, Lancet Public Health, 2024). We therefore added two figures to the Supplementary materials (Figure S5 and S6) showing the age and spatial distribution of the data and the reference simulation set in each scenario (CPRD without waning, COVER without waning, CPRD with waning). We discuss the points where the simulations and the data differ, and how this can impact the results of the model in the Discussion:

“In the reference scenario without waning, the risk of primary vaccine failure estimated when fitting the model to the case data is 5.2% (95% Credible Interval (CI): 4.9%; 5.5%). In the early MMR2 scenario with no change in coverage, most vaccinated children aged 2 and 3 have received 2 doses of vaccine, leading to a reduction of vaccinated children with primary vaccine failure. This increased immunity in children aged 2 and 3 causes indirect protection to the rest of the population, leading to an overall 14.5% reduction of cases compared to the median number of cases in the reference simulations (IQR: -0.01%-26.03%). When waning

is included in the model, the risk of primary vaccine failure estimated by the model is lower (2.5%, 95% CI 2.2% ; 2.9%), and the impact of early MMR2 on overall case number is also slightly lower (12.25% (-2.26; 24.64)). It is difficult to assess which model is closer to the true impact of early MMR2 as both reference simulation sets slightly diverge from the data: the number of measles cases aged 2 and 3 was underestimated in the model with waning (315 (IQR: 266-373) cases, while 486 were observed in the data), hence it might underestimate the protective effect of bringing the second dose forward. The model without waning overestimated the number of vaccinated cases aged 2 and 3 (164 (IQR: 140; 193) cases, while 52 were observed in the data), hence it might overestimate the protective effect of an earlier second dose. We conclude that the number of cases in early MMR2 vaccination scenario was consistently 10% lower than the median number of cases in the reference simulations, and that the decrease was robust to changes in MMR2 coverage.”

Are improvements to cases similar across the different regions? (i.e., does London see the same relative gains in case reductions as regions with higher coverage like West Midlands?)

We added table to the supplementary material which breaks down the number of cases by region for the reference scenario, MMR1+ 0.5% and an early MMR2 scenario.

It shows slightly different patterns by vaccination strategy. If only the coverage of MMR1 was increased, the regions with the highest proportion of averted cases would be in the North East of England, Yorkshire and the Humber and the West Midlands. In contrast to this, when advancing MMR2, London would have the highest proportion of averted cases.

We added these interesting findings to the results section of the manuscript:

“Through either increasing the coverage of MMR1 or advancing MMR2, the years with the highest proportion of cases averted are 2013, 2014, and 2015 (see Table S5). When increasing MMR1 by 0.5%, the highest proportion of avoided cases was in the North East of England, Yorkshire and the Humber and the West Midlands, in contrast to an advanced MMR2 which reduced most measles cases in London (see table S6). However, the differences in proportion of cases averted per regions were small in both scenarios, with median proportion ranging between 14.9% and 17.6%.”

The majority of cases arise from 2011-2013. Given that the model is fixed at 2010 (all trajectories for all scenarios start at the same value) is it possible that the effect of the policies is underestimated? One way to explore this would be to evaluate the effect for a fixed year. Is the effect on measles consistent year by year?

Overall, the study illustrates the long-term impact of this policy change for a ten year period. To give more detail on the different models over time, we added a table

breaking down the number of annual cases. This table shows that 2013, 2014, and 2015 had the highest proportion of cases prevented. We have added these findings to the results section:

“Through either increasing the coverage of MMR1 or advancing MMR2, the years with the highest proportion of cases averted are 2013, 2014, and 2015 (see Table S5).

“

As we are using counterfactual scenarios which assume an immediate perfect implementation of the new vaccination schedule, our models are probably overestimating the effect of new policies as they are not allowing for a latency period when the new vaccination schedule is slowly adapted.

We are not exploring any long-term effects beyond the ten year period of the study.

Figure 1 shows a marked disjoint in coverage for both MMR1 and MMR2 between 2010 and 2011 for most regions. Is this due to the COVER data or can it be traced to a historical effect (e.g., change in policy, historical event, etc.). Is it possible to repeat the analysis only using the CPRD data? And then the COVER data? It also looks like it would be possible to estimate the region specific underascertainment of COVER based on the comparison with the CPRD data. Is there a reason not to do this and to prefer the 50% adjustment (from Suffel et al. 2023)?

Unfortunately, the analysis cannot be repeated using either COVER or CPRD data alone. This is also further explained in the response to comment #4 by the first reviewer. Both datasets must be used in a complementary way to obtain estimates for all younger age groups but also represent the immunity landscape for older generations in the model. We have added the following paragraph to the methods section:

“Two data sources of vaccine coverage were used: The Clinical Practice Research Datalink (CPRD) Aurum to estimate vaccine coverage by region and one-year age bands and the Cover of Vaccination Evaluated Rapidly (COVER) to supplement missing data for age groups not included in CPRD Aurum.”

CPRD Aurum was only available for children born from 2006 until 2015, as GP practices only started to adopt the EMIS software system in the early 2000 which is the base for CPRD Aurum. Due to the limited CPRD data available, the coverage estimates at the age of 5 for 2010 (children born in 2005) are derived from the COVER data but adjusted for underascertainment. We also used adjusted COVER data to estimate the vaccine coverage in older age groups and children born after 2015 as there was no younger cohorts available for the study. Adjusted COVER data alone

could have not been used for the model either as it only provides data for children aged 2 and 5. A smaller breakdown of these age groups was necessary to represent the changes in the vaccination schedule and associated changes in coverage for children between 2 and 5. This could be only achieved by the use of detailed patient-level electronic health record data, reflecting the changes of vaccination coverage for each year of age in younger children.

Figure 1 shows an outlier of the coverage estimate for 2011, whereas the supplemented value from COVER for 2010 aligns well with the rest of the CPRD data. The first year of data derived from CPRD (2011) was based on a smaller sample size than subsequent years.

In one of the sensitivity analyses, we fitting the model using the COVER data, supplemented with CPRD data to infer the proportion of children gaining vaccination at age 3 and 4 years of age (see supplementary section S5). The overall results were similar to the model using CPRD, however, the impact of an earlier MMR2 on reducing cases was smaller than in the models using CPRD data which can be explained by the overall lower vaccination coverage in the COVER data where outbreaks are more driven by unvaccinated children than by children with primary vaccine-failure. However, if MMR2 is given early and taken up with same coverage as MMR1, we can still see significant reduction of cases (17.22%, IQR: 1.54; 29.7).

It is unclear from the text and figures whether the increase in MMR1 and MMR2 simulated in the scenarios is a relative or absolute gain and whether it is year-to-year or over the course of the simulation. Some regions see a decline in coverage from 2010-2019 so do these scenarios use the full historical trajectory and simply amend each year or do you fix 2010 and then increase from that starting point? Please clarify.

We used an absolute increase by 0.25/0.5/1% of coverage for each region across the years 2010-2019. We clarified this in the methods text describing the scenarios:

“We created scenarios in which the timing of MMR2 was not changed from the original schedule but the overall coverage of MMR1 or MMR2 was increased across all regions and years between 2010 and 2019. We implemented an absolute increase by 0.5 and 1% for either MMR1 or MMR2 in every region between 2010 and 2019. As coverage for MMR2 at the age of five is lower than for MMR1, we further included a scenario with increasing MMR2 by 3% or reducing coverage by up to 3%.”

“We explored two alternative scenarios with MMR2 recommended at the age of two instead of three years and four months. The first assumed the uptake would follow the same pattern of timeliness as the current MMR2 delivery, therefore one year and four months were subtracted from all the dates when MMR2 was received. New coverage estimates were calculated from these updated dates. In the second, we assumed that MMR2 recommended at the age of two would be taken up with the same speed as

MMR1, which is usually faster than MMR2 (33). In the new schedule recommended by the JCVI, MMR2 would be delivered at 18 months, but since our model is stratified by age groups (1 year-age bands until 6), we used 24 months in the simulations.

We also explored the impact of moving MMR2 recommendation to five years of age, assuming that the speed of uptake was similar to the reference MMR2 speed. New coverage estimates were calculated from the updated dates.

Finally, we looked at potential scenarios in which an earlier MMR2 would also influence coverage for MMR2. For this we explored an increase by 0.25, 0.5 and 1% and added a scenario in which the coverage for MMR1 and MMR2 were equal. Lastly, we added one scenario in which the coverage of MMR2 would be negatively impacted by bringing MMR2 forward in the schedule.”

It might also be helpful to overplot the trajectories in Figure 1.

Consider updating Figure 1 by (1) moving the year to the x-axis (improved clarity and black and white printing), (2) only including the 8 regions considered (simplifies the figure and avoids confusion with 10 subplots since the simulation is stratified by region), and (3) use the same y axis scale for MMR1 and MMR2.

We updated figure 1, which now shows coverage of MMR1 and MMR2 using the same x-axis and present the regions as facets next to each other which makes it easier to compare changes in uptake between doses across all regions and England in total.

The title of the talk suggests that the results here are relevant to low-incidence settings. However the analysis is purely focused on the UK setting without substantive discussion or work on how it might extend to other locations. It seems more appropriate to set the context in the title “How vaccine timing, timeliness, and coverage impact measles outbreaks in low incidence settings”. Further along this point, the analysis does not go into the timeliness (e.g., children getting their vaccine at a time different than the policy) and only the timing (e.g., what is the policy). It is confusing to have “timeliness” in the title. Lastly, the results look purely at number of cases and don’t have much to say about the nature (e.g., number or severity) of outbreaks. Having the word “outbreak” in the title is also somewhat confusing.

We included some aspects of timeliness in our analysis exploring a scenario when MMR2 was taken up with the same speed as the first dose (see Table S8). However, we agree with the reviewer that this has not been the main focus of the paper and changed the title to : **“Impact of vaccination timing and coverage on measles near elimination dynamics: a mathematical modelling analysis”**

Although we agree that the simulation analysis is applied to the dynamics observed in England, we incorporate vaccine schedules from other low-incidence settings, and provide recommendations that can apply to other near elimination settings (but not to endemic settings). We have changed the title to “Impact of vaccination timing and coverage on measles near elimination dynamics: a mathematical modelling analysis”

We also reflect the transferability of our findings in the discussion:

“More work is needed to understand how vaccination schedules influences outbreak risks in near elimination countries: Vaccination schedules differ in near elimination countries, but differences in historical coverage, previous incidence, age structure of the population, and spatial distribution of coverage make direct comparison challenging. With the increase in measles cases observed across most European countries in 2023 and 2024, it is important to understand whether early MMR2 schedule is consistently associated with lower outbreak risk (39).”

Why is the github repository “measles_england_sir” when the model is an SEIR model? The repository is also a branch on the repository for Robert et al. 2024. I’d recommend making a separate repository to avoid confusion.

We followed the reviewer’s suggestion and created a separate repository which can be found under the following link:

https://github.com/Eyedeet/measles_vaccination_scenarios

For their analysis, Roberts et al. used 5k samples per scenario. Why does this analysis only use 250?

We used 2500 simulations to create our scenarios. We explored different numbers of simulations and the estimates did not change significantly after more simulations. Hence, as a compromise between number of simulations and available working memory to create simulations, we finally remained with 2500 simulations.

We thank the reviewer for their constructive comments. We respond to each comment point by point.

As we tested additional analyses suggested by the reviewer (e.g. moving the interventions to 2015), we fixed an error in the implementation of our models to avoid individuals moving from Susceptible to “Vaccinated twice” within 1 year. These fixes led to an update of our results, and some minor changes as waning now reduces the impact of early delivery of MMR2 (from 12% decrease in the reference scenario, to 5.3% decrease when waning was included in the model). Moving MMR2 forward is still consistently associated with a decrease in case numbers throughout the scenarios, but we updated the results and discussion to reflect on the tradeoffs associated with waning of vaccine-induced immunity.

Following this, two authors (AS & AR) have checked all code files and datasets again. We also re-ran all analyses and outputs to check for further inconsistencies.

I was not able to find the estimated cases for the calibration split out by region (either in this paper or the original paper by Roberts et al.). The fit to the age data is documented but the regional dynamics is not.

We present the regional breakdown of measles cases in the supplementary material Figure S1.D, and the comparison between the reference scenario and the spatial data in Figure S5 (panels C and E). The spatial calibration is also described in the original paper (“Discrepancies in the spatial distribution of cases were expected because R_0 was not stratified by region, and spatial heterogeneity in transmission risk only depended on region-stratified vaccine coverage, available for cases vaccinated from 2004 onwards”). Following the first round of revisions, we also added a table to the supplementary material which shows the number of measles cases by region and vaccination scenario (see Table S6).

While Figure 1 was updated following specific suggestions, they remain overall in need of improvement. For example:

- case time series should have a y-axis that starts at 0 and share common y axis (particularly since the ranges are similar)
- all labels should include information about units
- figures should have had appropriate resolution (Fig 2 and Fig 3).

In my opinion, the quality of the figures needs to be improved for publication in a journal such as nature communications.

I’m convinced there is a better way to plot this data. Having 10 subplots take up an entire page is not an efficient use of a figure.

All case time series have now the same y-axis which starts at 0 (see figure 4, 5, and supplementary figures). All labels include the units.

We further improved the resolution of Figure 2 & 3.

We have edited figure 1, which now combines uptake for MMR1 and MMR2 for each region in the same plot and reduced the number of subplots to four plots.

The new repository is helpful. However, there it is missing a few basic practices that are essential for reproducibility (e.g., package versions, lockfile). This is particularly important since there are limitations on the version of R (e.g., dust requires R>4.0.0). This might help with some potential issues with the repository. For example:

- excel.link is listed as a requirement, but never utilized (it is also only available for Windows).

The excel.link function is used in the seirvodin package to read in the population estimates and the case data from excel sheets of the original data. As this is not necessary for using the fitted model, we removed the package from the script.

- Missing library “cowplot”

The library “cowplot” is called in the all_figures.R script. We added it to the README file.

- Using curly quotes for instructions to run scripts rather than straight quotes (e.g., source(“R/Outbreak_scenarios_CPRD.R”) vs source("R/Outbreak_scenarios_CPRD.R")) - No packages loaded for "R/all_figures.R".

The “cowplot” package is called in the all_figures.R script. To improve readability, we moved the package to the top of the script. All other packages required are already loaded through the “function_figures.R” script but also added to the README file.

We replaced the curly brackets quotes with straight quotes (') in the README file.

Additionally, there is some information in the repository that is wrong. For example:

- No standard laptop has 128GB of RAM. Knowing the number of cores would be relevant given that there are timings reported.

We used a virtual machine with 128GB of RAM to run simulations and made this more explicit in the description of the GitHub repository.

The description now reads as “The overall runtime does not exceed 37 hours for running 16 scenarios per script on a virtual machine with a 32-Core 3.0 GHz processor and 128 GB RAM.”

- Typo in the name of the script: source(“R/Outbreak_sencarios_CPRD.R”)

We corrected the typo.

General comments:

I appreciate the authors’ responses and updates to the manuscript; the model and analysis are clearer. I also appreciate the new repository which is documented and organized. I’d encourage a few changes to improve the repository for reproducibility. I’ll also note that the study notes that some of the analyses is limited by simulations being “infeasible” to run. Based on the github it seems like the analyses were being run on a

laptop. It seems like switching to a VM could have easily enabled larger sampling sets. It looks like seirvodin does not use the multi[1]threading capability of odin_dust which would provide an immediate speedup by ~xnumber of cores. Right now, it looks like the work is restricted to a single core. My expectation is that the constraint has to do with the data rather than compute.

Along these lines, it is surprising that the calibrated model is not presented by region (either in this The authors note that they explored larger sampling sizes and it did not affect the results – this is the relevant point to note. WRT increasing spatial resolution this also seems like it should be computationally feasible. What is relevant is whether it is necessary for the study and if the data allows it – this does not appear to be discussed.

We thank the author for this comment. Indeed, the analysis was run on a virtual machine with a higher working memory than a normal laptop (128GB). The regional breakdown of the estimated cases is already presented in the supplementary material (Table S6), along with a comparison between the reference simulations and the data (Figure S5, panels C and E).

The only feasibility issue mentioned in the study relates to the integration of more detailed spatial granularity. We agree with the reviewer that this issue is partly linked to data availability: Adding more spatial granularity to the model would require using the upper tier local authority level in England (instead of regions), the detailed vaccination data for the younger age groups is only available on a regional level to preserve anonymity of the included individuals. Furthermore, the vaccination data of older age groups is not consistently available with that level of detail (especially in individuals born prior to 2010).

Using the upper tier local authority level in England instead of regions also comes with computational issues. It would mean moving from nine regions to more than 100 local authorities. As each spatial unit is stratified in 12 age groups, and each stratum is made of 12 compartments (Figure S4), this would result in a model with more than 10,000 compartments, which would need to be re-fitted and would be significantly slower than the current implementation. Multithreading may alleviate some of that burden, but we believe that the computational cost should be acknowledged.

In agreement with the reviewer's comment, we have edited the limitation section related to spatial granularity:

“Increasing the spatial granularity of the model would mean using upper tier local authorities instead of regions and would lead to two main issues: CPRD vaccine data was not available on a more granular scale to protect anonymity. Secondly, increasing spatial granularity would multiply the number of strata in the model: there are nine regions in England, but more than 100 local authorities, which would greatly increase the number of compartments in the model (100 spatial units, 12 age groups and 12 compartments per strata would result in more than 10,000 compartments). This would slow down fitting the model and generating the stochastic simulations.”

My main concern about the generalizability of the main result (e.g., whether “early” MMR2 is better than the original schedule) is that the analysis rests on a very specific scenario/location + counterfactuals. Furthermore, the analysis doesn’t provide a perspective on when the original scenario might be preferable (if any). Alternate scenarios explored (early MMR2 + reduced coverage or early MMR2 + early waning) demonstrate that a trade off exists but does not provide guidance on how to navigate that tradeoff – outside of additional modeling. Furthermore, all results were extracted for a specific range of infection history in England. In this case the decade considered had a large outbreak at the beginning and then notable drops after. As a result, I suspect that if, for example, the modeling assumed that the intervention started at 2015 and looked at resulting case data then the estimated difference would be quite smaller than what is currently reported. In fact, due to the size of the outbreak I’m not convinced that the size of the impact is not essentially arguing for an age targeted campaign timed for that outbreak (early ages being more effective than later ages) rather than a change to the RI system. It seems to me that the reported 15% is quite welded to the specific scenario. This, in my opinion, makes the conclusions of the analysis, while an interesting case study, not as immediately useful to help policy in other near-elimination settings. I am concerned that the language is too strong to suggest so and that the general audience of the journal would assume just that

We thank the reviewer for this comment. As highlighted at the start of this response, we have added a fix to the models, which slightly affected the impact of waning. We split this comment into different sections and answer them specifically.

Generalisability of the findings to other near-elimination settings:

Our study is focused on England as an example of a low-incidence setting, and we agree that there are setting-specific circumstances that may impact the effect of changing timing. To avoid any misunderstanding from the readers, we have edited the end of the abstract to clarify that the results and conclusions were based on data from England:

“While larger increase in first-dose coverage provided the best outcomes, it may not be achievable. Thus, an earlier second MMR dose is a feasible alternative, and led to a decrease in the size of measles outbreaks simulated in England.”

We also edited the Discussion to further explain the limitations of transferability of our findings to other low-incidence countries, and highlight that more epidemiological analyses would be needed to fully evaluate the impact of vaccine schedule on transmission dynamics:

“More work is needed to understand how vaccination schedules influences outbreak risks in near elimination countries: Vaccination schedules differ in near elimination countries, but differences in historical coverage, previous incidence, age structure of the population, and spatial distribution of coverage make direct comparison challenging. With the increase in measles cases observed across most European countries in 2023 and 2024 (39), it is important to understand whether early MMR2 schedule is consistently associated with lower outbreak risk (40). Further epidemiological or mathematical modelling analysis using historical case and vaccination data from near-elimination countries would be needed to fully understand how our findings can be transferred to other settings.”

However, we think that these results are still useful to other low-incidence settings as they rely on ten years of data, which includes multiple outbreaks with different transmission patterns (2011-2013, and 2018-2019). Although the proportion of averted cases varied depending on the age groups affected by the outbreak (between 6% and 17% per year), moving to early delivery consistently led to a reduction in the median number of cases.

Lack of tradeoffs around the change in schedule:

As highlighted by the reviewer, the article explores various tradeoffs around the change in schedule, mostly around the impact of reduced MMR2 coverage and waning. We highlight that 1/ reduced MMR2 coverage did not outweigh the impact of early MMR2 delivery (“Changes in the vaccine schedule could lead to significant short-term improvements and may mitigate risks of large outbreaks. This decrease was observed even if changing the vaccine schedule led to decreasing MMR2 coverage by up to 3%.”). and 2/ early waning of immunity slightly reduced the benefits of early MMR2 delivery, but the number of cases remained lower than in the reference scenario (“In a sensitivity analysis, we showed that the median number of cases with early MMR2 vaccination remained 9% smaller than the reference simulations when waning started at 3 years instead of 5. This shows that the short- and medium-term consequences of waning do not fully outweigh the impact of early MMR2 vaccination in our model.”). Therefore, all scenarios and sensitivity showed that early delivery of MMR2 led to a small to moderate improvement, and this is reflected in the conclusions.

However, we also mention elements that should remain monitored (besides mathematical modelling) to evaluate the effect of early delivery: for instance, the MMR2 coverage following a change in vaccination appointment (“Determinants of childhood vaccine uptake are complex, and include parental vaccine confidence, opportunities to ask questions, reminders and barriers to vaccination appointments, which could all be impacted by changes in vaccine schedules (35). However, vaccine uptake is generally higher for the earlier appointments in the UK childhood vaccination schedule and so it may be reasonable to assume that an earlier MMR2 date is likely to result in higher uptake (33). Introducing an earlier second MMR2 at a similar time to a routine health visit for the child might also offer additional opportunity to remind parents of the vaccination and provide information.”). We also highlight the importance of maintaining a close monitoring of cases in vaccinated individuals, as early waning of vaccine-induced immunity could reduce the impact of change in delivery. Finally, although this is not a direct example of tradeoff, we also show what increase in MMR1 would lead to benefits similar to early delivery of MMR2, which indicates what is needed to mitigate outbreak risks under the current policy.

Later start of interventions:

Following the mention of a start of interventions in 2015 by the reviewers, we thought it would be interesting to look into the effect of delayed start of interventions on case numbers. When the early MMR2 vaccination started in 2015, the proportion of cases avoided in 2018-2019 was 6.65 (-14.37; 24.98), in line with the reference simulations (7.4% cases avoided in 2018-2019). However, the proportion of cases avoided in 2015-2017 was lower than in the reference simulations (5.9% when interventions start in 2015, 10.5% in the reference simulations). This difference is due to the transition period until 2018, by which all children have received an early second dose.

We have clarified that the main analysis does not take into account this transition period in the "changing MMR2 schedule" subsection in the Methods:

"We assumed that all vaccinated children had received their second dose at 2 at the start of the simulations. This means that the effects of a transition period where part of the children would still be vaccinated on the original schedule are not considered in these scenarios."

REVIEWER 3 full report:

**** Summary ****

This paper assesses the impact of various policies for the recommended age of measles vaccination in England (specifically the second dose, MMR2). They utilize a compartmental model of measles transmission fit to case data from 2010-2019 stratified by the 8 regions in England. The authors conduct counterfactual analyses assessing how cases would have changed for a suite of vaccination coverage and vaccination age policies for MMR2. The analysis is relevant given the recent JCVI recommendation to bring MMR2 forward to 18 months by 2025.

The paper is promising and well-constructed. However, it is missing some critical information regarding the model without which it is difficult to assess its validity and relevance to the question at hand.

**** Strengths: ****

The analysis is timely given the JCVI recommendation to bring MMR2 forward to 18 months. The JCVI recommendation was motivated by evidence that this might increase coverage; this analysis also suggests that earlier timing (18 month MMR2 recommendation) may also reduce overall cases.

The results include supporting studies including waning and school entry scenarios.

The authors utilize two sources to estimate vaccination coverage. Most of the time span is informed by electronic health records (CPRD). However, for times when the records are not available the authors utilize summaries from NHS Digital (COVER).

Weaknesses:

The text is missing some information regarding calibration and model relevance. The model appears to be lifted from another recent study by the same authors (Roberts et al. 2024). However, the paper under consideration does not provide the details necessary to assess the model (and consequently the analysis) directly from the text. In my opinion it is not sufficient to reference another paper for essential model details such as parameters and calibration methodology. When I look at the paper by Roberts et al. 2024 it is not entirely clear which of the fitted models this analysis is using. Please see my comments to the authors.

We have added more information on the calibration and fitting of the model in the Supplement and the Main Paper, along with figures comparing the simulations and the case data. See specific answers to the comments made by the reviewer,

Some of the figures could be improved for clarity.

The figures were updated according to specific points made by the reviewers.

While Figure 1 was updated following specific suggestions, they remain overall in need of improvement. For example:

- **case time series should have a y-axis that starts at 0 and share common y axis (particularly since the ranges are similar)**
- **all labels should include information about units**
- **figures should have had appropriate resolution (Fig 2 and Fig 3).**

In my opinion, the quality of the figures needs to be improved for publication in a journal such as nature communications.

The analysis is purely counterfactual, focusing on recent historical cases. The study cannot comment on how future projections of demographic, vaccination trends, or case importation may impact potential measles cases with the considered policies.

This point has been added to the limitations of the paper *“Finally, this analysis focuses on counterfactual scenarios of measles transmission between 2010 and 2019, and does not account for the impact of vaccination trend, demographics, or global measles dynamics on future outbreaks in England”*.

It is hard to follow how yearly coverage at age 5 (e.g., figure 1) maps onto vaccination rates for the different age bins. This is particularly important since the authors are using 2 different data sources for their coverage data (CPRD and COVER).

We updated figure 1 to better illustrate the CPRD vaccine data by English region over time and we clarified the mapping of the coverage data to the age bins in the methods section, which now reads as:

“As the recommended age for MMR vaccination is one year (MMR 1) and three years and four months (MMR 2), most children receive a first dose between one and two, and a second dose between three and four. Therefore, the vaccination coverage for MMR1 at age 1 and MMR2 at age 3 is nearly zero in the data, but a large proportion of children aged one have actually received a dose of vaccine (same with MMR2 for children aged three). To adjust the coverage data in the 1-year age band structure of the model, we assumed the coverage of MMR1 at the age of 1 was 75% of the coverage for MM1 at age 2, and 50% of the coverage at age 3 of the coverage of MMR2 at the age 4.”

The study is limited to England and it is unclear how much is relevant to low incidence settings. The title in general seems to be somewhat misaligned with the focus of the paper. Please see my comments to the authors.

We changed the title of the paper according to the reviewer’s comment (see below) and added some more information on how these findings are more generalisable to other low-incidence settings (see response to comment below)

The study only briefly comments on the tradeoff of boosting effect w/ age (I337-338) and does not consider at all the potential reduction in outbreaks in school/pre-school settings. Without discussion of this the paper feels incomplete particularly w/ the evidence of waning from Roberts et al. (2024). Thinking more long term, the potential effect of boosting strength is potentially critical.

We have added an extra sensitivity analysis exploring the short and medium-term impact of early waning as a consequence of the change in vaccination schedule, and discussed its impact. However, this analysis relies on the assumption that the waning rate is not affected by the age of vaccination (i.e. waning starts early but is not faster), if changing the vaccination schedule impacted the waning rate, then the tradeoffs may be more balanced:

“We showed that the median number of cases with early MMR2 vaccination remained 9% smaller than the reference simulations when waning started at 3 years instead of 5. This shows that the short and medium term consequences of waning do not outweigh the impact of early MMR2 vaccination in our model. However, close monitoring of transmission in vaccinated adults must be maintained to understand long-term immunity of vaccinated individuals in countries near elimination.”

**** Comments to the authors ****

How is the model calibrated? The text says that details about the model fitting process can be found in the supplementary material (I150) but S5 does not appear to contain any information related to the model fitting process. - What data is being used for the fit? Is it just yearly case data by region? - What parameters are being fit? What are the fitted parameter values and do you have uncertainty estimates for them?

We added more detail on the model to the study design, clarifying that it has been fitted to daily cases by region, age and vaccination status.

“We generated stochastic simulations using a mechanistic transmission model that had previously been fitted to the daily number of confirmed cases reported in England stratified by age and region with cases by vaccination status (26).”

We further added more explanation to the description of the transmission model in the main text how the stochastic simulations were generated:

“A more detailed description of the model fitting process and model parameters can be found in the supplementary material (see S3). We ran 2500 simulations per vaccination strategy by drawing 100 parameter sets from the model fits and 25 simulations per parameter set. Parameters drawn from the model fits included infection rate, duration of maternal immunity, parameters for seasonality and importation, report rate of imported cases, vaccine effectiveness, existing immunity in older generations and parameters of spatial spread (see Table S7). In the stochastic simulations, the number of transitions was computed using binomial draws, with a rate of transition derived from the distribution of the population between compartments at the previous time step, and the parameter set. Counterfactual scenarios were matched by using the same seed for the stochastic simulations. “

We added more details to Supplementary Section S3, to explain how age and spatial contact rates are estimated in the model, we also listed the parameters of the model and presented the model estimates (see tables S2 and S3).

Thank you for the additional information.

I was not able to find the estimated cases for the calibration split out by region (either in this paper or the original paper by Roberts et al.). The fit to the age data is documented but the regional dynamics is not.

Does the model include any age mixing? If so, how is this informed or constrained?

Does the model include seasonality? If so, how is this informed or constrained?

Does the model include importations. If so, how is this informed or constrained?

The model included age mixing, seasonality parameters and importations. The age mixing was informed by the age-stratified contact matrix from the POLYMOD study. Within-year seasonality of transmission was informed by two parameters estimated by the model.

Importations are also included in the model: The average yearly number of importations per region was computed from the number of cases classified as “imported” or “import related” in the individual case data for each region and year. This local number of importations for each year is then divided by 365 and weighted by the number of inhabitants per age group, to get the daily importation rate by age group and region. We considered that importations are less likely to be reported than other cases, so the importation rate by region is divided by p_{import} , the probability of reporting of imported cases. The model also estimates two parameters X_{import} and Y_{import} , to estimate the within-year seasonality of importations. The model therefore estimates three parameters to compute the number of importations:

p_{import} , X_{import} and Y_{import} .

We added this information in the supplementary material (S3) which reads as:

“Contacts between age groups were included to the model based on the contact matrix from the POLYMOD study (3).

Contacts across the nine regions of England were approximated using a spatial kernel functions (see estimated parameters in table S7). This kernel is a gravity model, depending on population in both regions, and distance. Distance is accounted for by the

degree of connectivity between regions (neighbours have a degree of one, neighbours of neighbours have a degree of two etc.).

The within-year seasonality of transmission was estimated by two parameters (X and Y). For each time t , the infection rate (β) is computed as $\beta_t = \beta * (1 + X * \cos(\frac{2*\pi*t}{365.25} + Y))$.

The average number of importations per year and region was computed using the number of cases who were classified as “imported” or “import related” in the individual case data. The daily importation rate by age group and region was then computed by dividing the local number of importations for each year by 365 and weighted by the number of inhabitants per age group. As importations may be less likely to be reported than other cases, the importation rate by region is divided by p_{import} , the report rate of imported cases. The model also estimates two parameters X_{import} and Y_{import} , to estimate the seasonality of importations in a given year.

$$n_{import}(a, i, t) = \frac{n_{import}(i, year(t))}{365 * p_{import}} * \frac{N_{ai}}{N_i} * (1 + X_{import} * \cos(\frac{2 * \pi * t}{365.25} + Y_{import}))$$

With N_{ai} the number of inhabitants of age a in i , N_i the number of inhabitants in region i , $n_{import}(i, year(t))$ the number of cases classified as “imported” or “import related” in the individual case data at each year. (Mossong et al., 2008) (Mossong et al., 2008)

We further describe the estimated parameters in the new supplementary tables S2 and S3, and figure S7 which describes importations and seasonality in more detail.

The results are presented as mean and IQ values of cases from 250 evaluations of the model. Is the spread in model trajectories driven by the seed or do they also represent uncertainty from any of the model parameters (152153)?

We drew 100 parameter sets from the model estimates, and generated 25 simulations for each set, which amounts to 2,500 simulations in total. At each time step and for each compartment, the number of transitions between compartments is drawn from a binomial distribution, with the rate computed from the distribution of the population between compartments, and the parameter set. The spread in model trajectories comes from 1/ differences between the parameter sets, and 2/ stochasticity in binomial draws.

We have expanded on this in both the main text and the supplementary material, proving a clearer description which parameters are estimated and where the uncertainty of the simulations comes from.

“We ran 2500 simulations per vaccination strategy by drawing 100 parameter sets from the model fits and running 25 simulations per parameter set. Parameters drawn from the model fits included infection rate, duration of maternal immunity, parameters for seasonality of transmission and importation, report rate of imported cases, vaccine effectiveness, existing immunity in older generations and parameters of spatial spread (see Table S7). In the stochastic simulations, the number of transitions was computed using a binomial draw

Thank you for the clarified text.

The text says that scenarios are matched using “the same seed for the stochastic simulations”. However, in S5 the model is described as “deterministic”. These appear to be incompatible descriptions. Can the authors clarify?

We fitted a deterministic compartmental model in the first place. However, the simulations are stochastic, using different particles and drawing different sets of parameters for the different simulations.

This is also now clarified in the methods:

“We generated stochastic simulations using a mechanistic transmission model that had previously been fitted to the daily number of confirmed cases reported in England stratified by age, region, and vaccination status (26). We used the parameter estimates obtained from the deterministic model fits to simulate stochastic outbreaks between 2010 and 2019.”

Thank you for your clarification.

Since the analysis is focused on age at vaccination policy it is critical that the model is reproducing the age distribution of cases. Can the authors include model fits to the UK HAS data in addition to Figure 5?

Figure 3 of the manuscripts shows the comparison of the UKHSA case data (see dots) to the model fit of the reference scenario (CPRD data without waning). We agree with the reviewer that it is important to show the comparison between the stochastic simulations and the data, especially with regards to the age distribution of the cases, but also did not want to repeat too much of the content from our previous analysis (Robert et al, Lancet Public Health, 2024). We therefore added two figures to the Supplementary materials (Figure S5 and S6) showing the age and spatial distribution of the data and the reference simulation set in each scenario (CPRD without waning, COVER without waning, CPRD with waning). We discuss the points where the simulations and the data differ, and how this can impact the results of the model in the Discussion:

“In the reference scenario without waning, the risk of primary vaccine failure estimated when fitting the model to the case data is 5.2% (95% Credible Interval (CI): 4.9%; 5.5%). In the early MMR2 scenario with no change in coverage, most vaccinated children aged 2 and 3 have received 2 doses of vaccine, leading to a reduction of vaccinated children with primary vaccine failure. This increased immunity in children aged 2 and 3 causes indirect protection to the rest of the population, leading to an overall 14.5% reduction of cases compared to the median number of cases in the reference simulations (IQR: -0.01%-26.03%). When waning is included in the model, the risk of primary vaccine failure estimated by the model is lower (2.5%, 95% CI 2.2% ; 2.9%), and the impact of early MMR2 on overall case number is also slightly lower (12.25% (-2.26; 24.64)). It is difficult to assess which model is closer to the true impact of early MMR2 as both reference simulation sets slightly diverge from the data: the number of measles cases aged 2 and 3 was underestimated in the model with waning (315 (IQR: 266-373) cases, while 486 were observed in the data), hence it might underestimate the protective effect of bringing the second dose forward. The model without waning overestimated the number of vaccinated cases aged 2 and 3 (164 (IQR: 140; 193) cases, while 52 were observed in the data), hence it might overestimate the protective effect of an earlier second dose. We conclude that the number of cases in early MMR2 vaccination scenario was consistently 10% lower than the median number of cases in the reference simulations, and that the decrease was robust to changes in MMR2 coverage.”

Thank you for the expanded discussion.

Are improvements to cases similar across the different regions? (i.e., does London see the same relative gains in case reductions as regions with higher coverage like West Midlands?)

We added table to the supplementary material which breaks down the number of cases by region for the reference scenario, MMR1+ 0.5% and an early MMR2 scenario.

It shows slightly different patterns by vaccination strategy. If only the coverage of MMR1 was increased, the regions with the highest proportion of averted cases would be in the North East of England, Yorkshire and the Humber and the West Midlands. In contrast to this, when advancing MMR2, London would have the highest proportion of averted cases.

We added these interesting findings to the results section of the manuscript:

“Through either increasing the coverage of MMR1 or advancing MMR2, the years with the highest proportion of cases averted are 2013, 2014, and 2015 (see Table S5). When increasing MMR1 by 0.5%, the highest proportion of avoided cases was in the North East of England, Yorkshire and the Humber and the West Midlands, in contrast to an advanced MMR2 which reduced most measles cases in London (see table S6). However, the differences in proportion of cases averted per regions were small in both scenarios, with median proportion ranging between 14.9% and 17.6%.”

Thank you for the additional information.

The majority of cases arise from 2011-2013. Given that the model is fixed at 2010 (all trajectories for all scenarios start at the same value) is it possible that the effect of the policies is underestimated? One way to explore this would be to evaluate the effect for a fixed year. Is the effect on measles consistent year by year?

Overall, the study illustrates the long-term impact of this policy change for a ten year period. To give more detail on the different models over time, we added a table breaking down the number of annual cases. This table shows that 2013, 2014, and

2015 had the highest proportion of cases prevented. We have added these findings to the results section:

““Through either increasing the coverage of MMR1 or advancing MMR2, the years with the highest proportion of cases averted are 2013, 2014, and 2015 (see Table S5).

“

As we are using counterfactual scenarios which assume an immediate perfect implementation of the new vaccination schedule, our models are probably overestimating the effect of new policies as they are not allowing for a latency period when the new vaccination schedule is slowly adapted.

We are not exploring any long-term effects beyond the ten year period of the study.

This is helpful, thank you. I agree that your model is probably overestimating the effect of new policies. My assessment is that this may be, in part, due to the fixed selection of the ten year period of the study. Please see general comments at the end.

Figure 1 shows a marked disjoint in coverage for both MMR1 and MMR1 between 2010 and 2011 for most regions. Is this due to the COVER data or can it be traced to a historical effect (e.g., change in policy, historical event, etc.). Is it possible to repeat the analysis only using the CPRD data? And then the COVER data? It also

looks like it would be possible to estimate the region specific underascertainment of COVER based on the comparison with the CPRD data. Is there a reason not to do this and to prefer the 50% adjustment (from Suffel et al. 2023)?

Unfortunately, the analysis cannot be repeated using either COVER or CPRD data alone. This is also further explained in the response to comment #4 by the first reviewer. Both datasets must be used in a complementary way to obtain estimates for all younger age groups but also represent the immunity landscape for older generations in the model. We have added the following paragraph to the methods section:

“Two data sources of vaccine coverage were used: The Clinical Practice Research Datalink (CPRD) Aurum to estimate vaccine coverage by region and one-year age bands and the Cover of Vaccination Evaluated Rapidly (COVER) to supplement missing data for age groups not included in CPRD Aurum.”

CPRD Aurum was only available for children born from 2006 until 2015, as GP practices only started to adopt the EMIS software system in the early 2000 which is the base for CPRD Aurum. Due to the limited CPRD data available, the coverage estimates at the age of 5 for 2010 (children born in 2005) are derived from the COVER data but adjusted for underascertainment. We also used adjusted COVER data to estimate the vaccine coverage in older age groups and children born after 2015 as there was no younger cohorts available for the study. Adjusted COVER data alone could have not been used for the model either as it only provides data for children aged 2 and 5. A smaller breakdown of these age groups was necessary to represent the changes in the vaccination schedule and associated changes in coverage for children between 2 and 5. This could be only achieved by the use of detailed patientlevel electronic health record data, reflecting the changes of vaccination coverage for each year of age in younger children.

Figure 1 shows an outlier of the coverage estimate for 2011, whereas the supplemented value from COVER for 2010 aligns well with the rest of the CPRD data. The first year of data derived from CPRD (2011) was based on a smaller sample size than subsequent years.

In one of the sensitivity analyses, we fitting the model using the COVER data, supplemented with CPRD data to infer the proportion of children gaining vaccination at age 3 and 4 years of age (see supplementary section S5). The overall results were similar to the model using CPRD, however, the impact of an earlier MMR2 on reducing cases was smaller than in the models using CPRD data which can be explained by the overall lower vaccination coverage in the COVER data where outbreaks are more driven by unvaccinated children than by children with primary vaccine-failure. However,

if MMR2 is given early and taken up with same coverage as MMR1, we can still see significant reduction of cases (17.22%, IQR: 1.54; 29.7).

Thank you for the detailed response.

It is unclear from the text and figures whether the increase in MMR1 and MMR2 simulated in the scenarios is a relative or absolute gain and whether it is year-to-year or over the course of the simulation. Some regions see a decline in coverage from 2010-2019 so do these scenarios use the full historical trajectory and simply amend each year or do you fix 2010 and then increase from that starting point? Please clarify.

We used an absolute increase by 0.25/0.5/1% of coverage for each region across the years 2010-2019. We clarified this in the methods text describing the scenarios:

“We created scenarios in which the timing of MMR2 was not changed from the original schedule but the overall coverage of MMR1 or MMR2 was increased across all regions and years between 2010 and 2019. We implemented an absolute increase by 0.5 and 1% for either MMR1 or MMR2 in every region between 2010 and 2019. As coverage for MMR2 at the age of five is lower than for MMR1, we further included a scenario with increasing MMR2 by 3% or reducing coverage by up to 3%.”

“We explored two alternative scenarios with MMR2 recommended at the age of two instead of three years and four months. The first assumed the uptake would follow the same pattern of timeliness as the current MMR2 delivery, therefore one year and four months were subtracted from all the dates when MMR2 was received. New coverage estimates were calculated from these updated dates. In the second, we assumed that MMR2 recommended at the age of two would be taken up with the same speed as MMR1, which is usually faster than MMR2 (33). In the new schedule recommended by the JCVI, MMR2 would be delivered at 18 months, but since our model is stratified by age groups (1 year-age bands until 6), we used 24 months in the simulations.

We also explored the impact of moving MMR2 recommendation to five years of age, assuming that the speed of uptake was similar to the reference MMR2 speed. New coverage estimates were calculated from the updated dates.

Finally, we looked at potential scenarios in which an earlier MMR2 would also influence coverage for MMR2. For this we explored an increase by 0.25, 0.5 and 1% and added a scenario in which the coverage for MMR1 and MMR2 were equal. Lastly, we added one

scenario in which the coverage of MMR2 would be negatively impacted by bringing MMR2 forward in the schedule.”

Thank you for your clarification.

It might also be helpful to overplot the trajectories in Figure 1.

Consider updating Figure 1 by (1) moving the year to the x-axis (improved clarity and black and white printing), (2) only including the 8 regions considered (simplifies the figure and avoids confusion with 10 subplots since the simulation is stratified by region), and (3) use the same y axis scale for MMR1 and MMR2.

We updated figure 1, which now shows coverage of MMR1 and MMR2 using the same x-axis and present the regions as facets next to each other which makes it easier to compare changes in uptake between doses across all regions and England in total.

I'm convinced there is a better way to plot this data. Having 10 subplots take up an entire page is not an efficient use of a figure.

The title of the talk suggests that the results here are relevant to low-incidence settings. However the analysis is purely focused on the UK setting without substantive discussion or work on how it might extend to other locations. It seems more appropriate to set the context in the title “How vaccine timing, timeliness, and coverage impact measles outbreaks in low incidence settings”. Further along this point, the analysis does not go into the timeliness (e.g., children getting their vaccine at a time different than the policy) and only the timing (e.g., what is the policy). It is confusing to have “timeliness” in the title. Lastly, the results look purely at number of cases and don't have much to say about the nature (e.g., number or severity) of outbreaks. Having the word “outbreak” in the title is also somewhat confusing.

We included some aspects of timeliness in our analysis exploring a scenario when MMR2 was taken up with the same speed as the first dose (see Table S8). However, we agree with the reviewer that this has not been the main focus of the paper and changed the title to : **“Impact of vaccination timing and coverage on measles near elimination dynamics: a mathematical modelling analysis”**

Although we agree that the simulation analysis is applied to the dynamics observed in England, we incorporate vaccine schedules from other low-incidence settings, and provide recommendations that can apply to other near elimination settings (but not to endemic settings). We have changed the title to “Impact of vaccination timing and coverage on measles near elimination dynamics: a mathematical modelling analysis” We also reflect the transferability of our findings in the discussion:

“More work is needed to understand how vaccination schedules influences outbreak risks in near elimination countries: Vaccination schedules differ in near elimination countries, but differences in historical coverage, previous incidence, age structure of the population, and spatial distribution of coverage make direct comparison challenging. With the increase in measles cases observed across most European countries in 2023 and 2024, it is important to understand whether early MMR2 schedule is consistently associated with lower outbreak risk (39).”

The talk title is a considered improvement – thank you. However, I think I have some concerns about the generalizability of these results to other settings. Please see general comments at the end of the document.

Why is the github repository “measles_england_sir” when the model is an SEIR model? The repository is also a branch on the repository for Robert et al. 2024. I’d recommend making a separate repository to avoid confusion.

We followed the reviewer’s suggestion and created a separate repository which can be found under the following link:

https://github.com/Eyedeet/measles_vaccination_scenarios

The new repository is helpful. However, there it is missing a few basic practices that are essential for reproducibility (e.g., package versions, lockfile). This is particularly important since there are limitations on the version of R (e.g., dust requires R>4.0.0).

This might help with some potential issues with the repository. For example:

- excel.link is listed as a requirement, but never utilized (it is also only available for Windows).
- Missing library “cowplot”
- Using curly quotes for instructions to run scripts rather than straight quotes (e.g., source(“R/Outbreak_scenarios_CPRD.R”) vs source("R/Outbreak_scenarios_CPRD.R"))
- No packages loaded for "R/all_figures.R".

Additionally, there is some information in the repository that is wrong. For example:

- No standard laptop has 128GB of RAM. Knowing the number of cores would be relevant given that there are timings reported.
- Typo in the name of the script: source(“R/Outbreak_sencarios_CPRD.R”)

For their analysis, Roberts et al. used 5k samples per scenario. Why does this analysis only use 250?

We used 2500 simulations to create our scenarios. We explored different numbers of simulations and the estimates did not change significantly after more simulations. Hence, as a compromise between number of simulations and available working memory to create simulations, we finally remained with 2500 simulations.

Thank you for the clarification and correcting the text.

General comments:

I appreciate the authors' responses and updates to the manuscript; the model and analysis are clearer. I also appreciate the new repository which is documented and organized. I'd encourage a few changes to improve the repository for reproducibility.

I'll also note that the study notes that some of the analyses is limited by simulations being "infeasible" to run. Based on the github it seems like the analyses were being run on a laptop. It seems like switching to a VM could have easily enabled larger sampling sets. It looks like seirvodin does not use the multi-threading capability of odin_dust which would provide an immediate speedup by ~xnumber of cores. Right now, it looks like the work is restricted to a single core. My expectation is that the constraint has to do with the data rather than compute. Along these lines, it is surprising that the calibrated model is not presented by region (either in this

The authors note that they explored larger sampling sizes and it did not affect the results – this is the relevant point to note. WRT increasing spatial resolution this also seems like it should be computationally feasible. What is relevant is whether it is necessary for the study and if the data allows it – this does not appear to be discussed.

My main concern about the generalizability of the main result (e.g., whether "early" MMR2 is better than the original schedule) is that the analysis rests on a very specific scenario/location + counterfactuals. Furthermore, the analysis doesn't provide a perspective on when the original scenario might be preferable (if any). Alternate scenarios explored (early MMR2 + reduced coverage or early MMR2 + early waning) demonstrate that a trade off exists but does not provide guidance on how to navigate that tradeoff – outside of additional modeling.

Furthermore, all results were extracted for a specific range of infection history in England. In this case the decade considered had a large outbreak at the beginning and then notable drops after. As a result, I suspect that if, for example,

the modeling assumed that the intervention started at 2015 and looked at resulting case data then the estimated difference would be quite smaller than what is currently reported. In fact, due to the size of the outbreak I'm not convinced that the size of the impact is not essentially arguing for an age targeted campaign timed for that outbreak (early ages being more effective than later ages) rather than a change to the RI system. It seems to me that the reported 15% is quite welded to the specific scenario. This, in my opinion, makes the conclusions of the analysis, while an interesting case study, not as immediately useful to help policy in other near-elimination settings. I am concerned that the language is too strong to suggest so and that the general audience of the journal would assume just that.